# Anion-exchange chromatography mass spectrometry provides extensive coverage of primary metabolic pathways revealing altered metabolism in IDH1 mutant cells

John Walsby-Tickle [1], Joan Gannon[1], Ingvild Hvinden [1], Chiara Bardella[2], Martine I. Abboud [1], Areesha Nazeer[1], David Hauton[1], Elisabete Pires[1], Tom Cadoux-Hudson[1], Christopher J. Schofield [1] & James S. O. McCullagh [1✉]

Altered central carbon metabolism is a hallmark of many diseases including diabetes, obesity, heart disease and cancer. Identifying metabolic changes will open opportunities for better understanding aetiological processes and identifying new diagnostic, prognostic, and therapeutic targets. Comprehensive and robust analysis of primary metabolic pathways in cells, tissues and bio-fluids, remains technically challenging. We report on the development and validation of a highly reproducible and robust untargeted method using anion-exchange tandem mass spectrometry (IC-MS) that enables analysis of 431 metabolites, providing detailed coverage of central carbon metabolism. We apply the method in an untargeted, discovery-driven workflow to investigate the metabolic effects of isocitrate dehydrogenase 1 (IDH1) mutations in glioblastoma cells. IC-MS provides comprehensive coverage of central metabolic pathways revealing significant elevation of 2-hydroxyglutarate and depletion of 2-oxoglutarate. Further analysis of the data reveals depletion in additional metabolites including previously unrecognised changes in lysine and tryptophan metabolism.

[1] Department of Chemistry, University of Oxford, Mansfield Road, Oxford OX1 3TA, UK. [2] Institute of Cancer and Genomic Sciences, University of Birmingham, Edgbaston, Birmingham B15 2TT, UK. ✉email: james.mccullagh@chem.ox.ac.uk

Metabolism associated with energy transduction has been fundamentally conserved during evolution; perturbations in central metabolic pathways are a hallmark of major non-infectious diseases, including diabetes, obesity, heart disease and cancer[1]. One or more of these diseases now affects a large fraction of the global population at some point in their lives. A detailed view of underlying metabolic changes in non-infectious diseases is needed to understand mechanisms of aetiology and find new diagnostic and prognostic markers to help develop more effective therapies[2–4]. Mass spectrometry (MS)-based metabolomics, using 'hyphenated' techniques (targeted and untargeted), has emerged as an efficient, sensitive and highly selective approach for in-depth, quantitative and comprehensive analysis of altered metabolic states. However, analysing metabolic changes in biological systems, which integrate comprehensively at the level of metabolic pathways, remains a technical challenge due to a lack of comprehensive compound coverage and identification[5]. The majority of metabolites associated with primary metabolic pathways, including those involved in energy transduction, nucleic acid metabolism, carbohydrate metabolism, redox metabolism and extended anaplerotic and cataplerotic processes, are acidic and thus predominantly in their negatively charged forms at physiologically relevant pH values[6]. Hydrophilic interaction liquid chromatography-MS (HILIC-MS), ion pair-MS (IP-MS) and derivatized gas chromatography-MS (GC-MS) are often used for the analysis of highly polar and ionic metabolites, but none are optimal for providing comprehensive, robust and reproducible coverage, particularly across central carbon metabolism[7,8]. Therefore, untargeted metabolomics applications, and functional interpretation of central metabolic states, are limited by current analytical capabilities and new, robust analytical methods, capable of enhanced pathway coverage and metabolite identification, can make an important contribution. Here we demonstrate that anion-exchange chromatography coupled to high-resolution orbitrap MS (IC-MS) provides comprehensive coverage of metabolites found in central carbon metabolism and a wide range of additional ionic metabolites.

IC-MS couples conventional ion-exchange chromatography with MS. Separation of compounds is based on ionic interaction between functional groups on a resin-based stationary phase and analytes in the mobile phase. Elution occurs by analyte exchange with higher ion-strength mobile phase ions, typically a gradient of hydroxide ions (for anion-exchange) or protons (for cation-exchange). Compatibility with MS is enabled by an ion suppressor placed between the ion chromatography system and the electrospray ion source of the mass spectrometer. As the chromatographic eluent passes through the suppressor, electrochemical conversion of hydroxide ions into water molecules takes place (in anion-exchange mode) or protons to water (in cation-exchange mode). This provides an electrospray-compatible chromatographic eluent that can be analysed directly over time by the mass spectrometer. The electrochemical ion-supresssor therefore removes hydroxide ions and oppositely charged counter ions, with both processes minimising potential ion suppression during electrospray ionisation in the ion source.

Proof of principle that IC-MS can be used for the analysis of metabolites has been reported previously, but its scope and application, particularly for comprehensive (untargeted) analysis of central carbon metabolism, has not been explored in detail[9–13]. A promising study by Schwaiger et al.[12] used 45 stable isotope-labelled metabolite internal standards for untargeted analysis of cell extracts and demonstrated high linearity, reproducibility, sensitivity and lower limit of detection (LLOD), but a comprehensive investigation of metabolites, pathway coverage and applicability to a wider range of tissues, cell and bio-fluid samples remains to be explored. We report the development and

application of a bespoke IC-MS method for analysis of anionic metabolites and its evaluation using a relatively large number of metabolite standards and range of sample types. We describe a workflow that integrates sample preparation and analysis enabling comprehensive identification of metabolites associated with interconnected central metabolic pathways in cells.

## Results

**Method development.** We first developed an analytical IC-MS method that balanced reproducibility, stability and coverage of the ionic metabolome with a minimised sample run time. This was tested on extracts from a wide range of sample types, including cells, tissues and bio-fluids. In order to validate this method for untargeted metabolomics applications, a protocol involving sample preparation through to MS electrospray ionisation, higher-energy collisional dissociation (HCD) fragmentation and detection, was created (see 'Methods').

A range of sample preparation protocols were assessed. It was found that in general both fully aqueous and fully organic sample diluents (including mixtures) were suitable for analysis and 80% (v/v) MeOH in Milli-Q purified water provided the highest recovery of selected ionic metabolites from cells and tissues. Of particular note was the fact that removal of soluble protein was shown to be essential to maximise ion-suppressor (Fig. 1a) lifetime (from a few months with MeOH solvent precipitation to >1 year with molecular weight cut-off (MwCO) filtration). We experimented with MwCO filtration of metabolite extractions from cells and found that 10 kDa MwCO filtration worked efficiently. No detrimental effect on recovery of ionic or highly polar metabolites was observed with use of MwCO filters applied to cell, tissues and blood plasma extracts; however, some reproducible loss (and in some cases enhancement) of lipid abundance and more hydrophobic compounds was observed (Supplementary Fig. 1a shows the effect of MwCO filtration on selected metabolite abundances). Optimisation of the ionisation source conditions in the mass spectrometer was explored to minimise in-source fragmentation and maximise ionisation efficiency. In addition, HCD fragmentation parameters were optimised to enable a range of metabolites to be automatically fragmented using a 'top 10' data-directed fragmentation acquisition (DDA) protocol (see 'Methods').

**Method validation.** We used a range of metabolite standards with differing physiochemical properties as well as metabolites from cell extracts for the determination of linearity, retention time reproducibility (robustness) and LLOD. Results from the targeted validation are given in Supplementary Fig. 1c, 1d. Of particular note are the highly reproducible retention times (Fig. 1b) compared to HILIC-MS analysis of the same biological samples (with both chromatography systems coupled to the same mass spectrometer). We evaluated untargeted method performance using cells, tissues and bio-fluids by comparing IC-MS analysis with standard reversed-phase-MS (RP-MS) and HILIC-MS[14]. Two different cell types (LN18 glioma cells genetically modified by lentiviral vector transduction to overexpress the R132H mutant form of isocitrate dehydrogenase 1 (*IDH1*) and untransduced *IDH1* wild-type cells) were grown to 90% confluency at two different glucose concentrations (5 and 25 mM glucose) providing four biological groups (quality control (QC) samples were comprised of an equal-volume mixture of all four experimental group samples). A principal component analysis (PCA) plot of all compound features by IC-MS is shown in Fig. 1c. QC samples plotted centrally and clustered tightly with separation of each of the four experimental groups. Both the RP-MS and HILIC-MS data provide a more diffuse grouping (Fig. 1c). These data demonstrate the reproducibility of

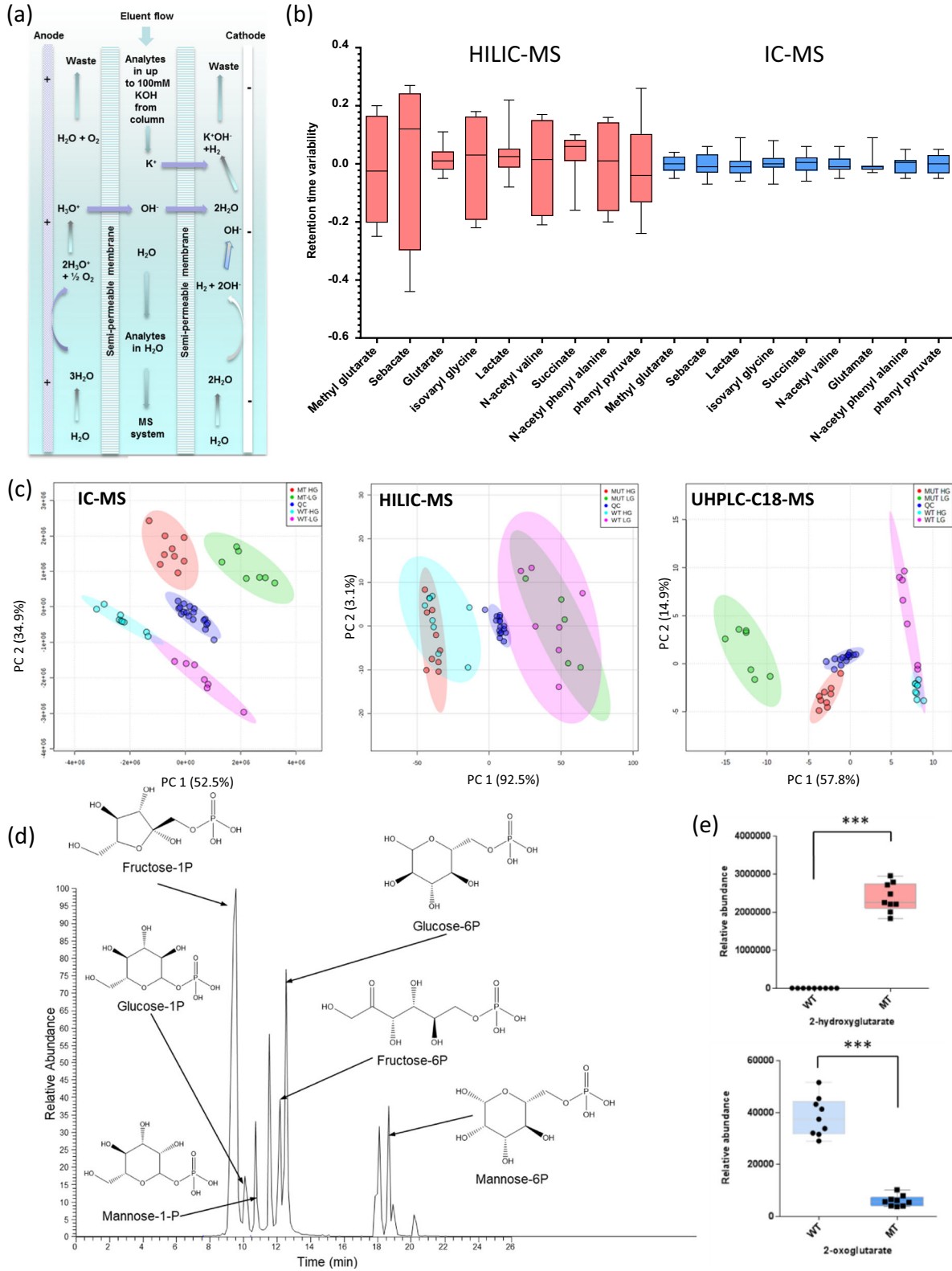

untargeted IC-MS analysis and reveal clear differences in metabolic composition between the biological groups analysed. It was noted that a change in glucose concentration had a major effect on the ionic metabolite profile. Supplementary Fig. 1e demonstrates that the number of compound features measured is higher for IC-MS compared to HILIC-MS for analysis of the same cell extracts; however, the number of compounds with %CV <15 and <30 were similar. Supplementary Figs. 1f, g provide untargeted

method validation results, including demonstration that IC-MS analysis was particularly sensitive to metabolic differences, as represented by the number of compound features measured and the tight clustering according to sample type.

**Extending ionic metabolome coverage**. To date IC-MS coverage of identified metabolites found in biological samples has been

**Fig. 1 Analysis of structural isomers and phosphorylated metabolites. a** Schematic illustrating the electrochemical process involved in an anion suppressor system used for IC-MS analysis. **b** Comparing retention time reproducibility (min) for HILIC-MS (red) and IC-MS (blue) using a selection of polar metabolites ($n = 18$ per metabolite; box extends from the 25th to the 75th percentile with median line; whiskers are min to max with all data points shown). **c** Principal component analysis (PCA) of the compound features from untargeted analysis of two cell types (LN18 IDH mutant bearing and LN18 IDH wild type) grown at two different glucose concentrations (5 and 25 mM), using three separate LC-MS/MS methods (IC-MS, C18 reversed-phase-MS and HILIC-MS). The tighter the clustering within a method per experimental group, for example, IC-MS vs. HILIC-MS, the better able the method is at revealing differences between experimental groups when taking all compound features into consideration (all samples were normalised to total DNA content and data normalised by sum; no scaling of data was employed. **d** Extracted ion chromatograms at $m/z$ 259.02244 (within 5 p.p.m. range), which represent $[M-H]^-$ deprotonated hexose monophosphates from LN18 glioblastoma cells. Separation of multiple structural isomers of hexose phosphates enables identification and relative quantification by integrated peak area (as shown here) or peak height. **e** Box plots showing differences in integrated peak area representing metabolite abundance for 2-hydroxyglutarate (top) and 2-oxoglutarate (bottom) between LN18 *IDH1* mutant and IDH wild-type cells cultured at high glucose ($n = 9$/group; box extends from the 25th to the 75th percentile with median shown by line. Whiskers are min to max with all data points shown. ***$P$ value < 0.001).

limited to a relatively small number of compounds using authentic standards[9,12]. To expand this substantially, we analysed authentic metabolite standards representing a wide range of primary metabolic pathways. We built a database of over 400 metabolites found in cells, tissues and bio-fluids that could be reliably detected and quantified using IC-MS. To identify metabolites in biological sample extracts, we compared four principle measurement parameters associated with compound features with those of the authentic standards: retention time, accurate mass, isotope abundance profile and HCD fragmentation pattern (where measured). Supplementary Data 1 provides a list of the metabolites that were amenable to IC-MS analysis and successfully identified. This list extends considerably what has so far been demonstrated for the analysis of metabolites using anion IC-MS, demonstrating its potential for future untargeted metabolomics applications[7,9–13].

**Extending separation of structural isomers**. The ability to differentiate structural isomers is important for a comprehensive analysis of central metabolism. For example, over 80% of the metabolites found in glycolysis, pentose phosphate pathway (PPP) and the tricarboxylic acid (TCA) cycle have additional structural isomers associated with other parts of endogenous mammalian metabolism (based on chemical formula search of the Human Metabolome Database (HMDB)). Previous studies have demonstrated that anion-exchange chromatography can differentiate selected hexose phosphates[12]. We extended this further to additional structural isomers of hexose phosphates and additional anionic structural isomers. Figure 1d provides an example of the separation of hexose monophosphates from the analysis of LN18 glioblastoma cells. Supplementary Fig. 2 shows extracted ion chromatograms (EICs) demonstrating the chromatographic separation of additional structural isomers: AMP and dGMP; ATP and dGTP; acetoacetate and 2-ketobutyrate; glutarate and ethylmalonic acid; *N*-acetyl-methionine and *N*-formyl-ethionine; 4-hydroxybenzoic acid and 3-hydroxybenzoic acid. Anion-exchange chromatography separates a range of metabolites with the same elemental composition and differing structures. This coverage of structural isomers, combined with inherent retention time precision, demonstrates a particularly robust platform enabling retention time to be used as a reliable differentiating factor, providing greater confidence in metabolite identifications and leading to a capability for more comprehensive metabolic pathway coverage.

**Analysis of phosphorylated metabolites**. Multiply phosphorylated metabolites, such as ATP, ADP, GTP, dGTP, GDP, CTP, dCTP, CDP and similar, are susceptible to loss of $PO_4^{3-}$ during electrospray ionisation[15]. This has also been reported in the context of IC-MS previously[12]. We investigated this process by

carefully modifying ion-source temperatures, gas pressure and voltages for analysis of di- and tri-phosphorylated metabolites (e.g. ADP and ATP, other nucleotides and sugar phosphates). Some ion-source loss of $PO_4^{3-}$ from ATP and ADP was identified by the presence of ADP and AMP, respectively, in the EIC of ATP. However, ATP, ADP and AMP were also each chromatographically resolved by several minutes from one another and therefore the proportion of $PO_4^{3-}$ loss attributed to post-column effects could be distinguished from endogenously derived ADP and AMP in the sample. We investigated $PO_4^{3-}$ loss due to post-column effects at different concentrations in cell extracts. The results (Supplementary Fig. 1b) demonstrate that $PO_4^{3-}$ losses were linear with respect to concentration and therefore not detrimental to the estimation of relative abundances of endogenous levels in cell samples.

**Analysis of zwitterions and modified amino acids**. Anion-exchange chromatography retains negatively charged metabolites, which are eluted from the column by an increasing hydroxide ion gradient. Online eluent suppression (via the ion suppressor, Fig. 1a) removes hydroxide ions from the mobile phase eluent but also all counter ions, that is, those with a positive charge. Zwitterions, such as many amino acids, are a special case as they contain potentially both positive and negatively charged functional groups depending on sample pH. Many amino acids, for example, are removed from the eluent stream by ion suppression, but we show here this does not apply to all amino acids or many modified amino acids/amino acid derivatives such as *N*-acetylated amino acids and oxo-acids (produced by amino acid transamination). Many metabolites associated with amino acid degradation pathways are not zwitterionic and are subsequently well characterised. Supplementary Fig. 3a illustrates EICs for a selection of modified amino acids (from the analysis of LN18 glioblastoma cells). Together, the results demonstrate that IC-MS can be used to investigate biologically important aspects of amino acid metabolism, despite a lack of suitability for analysing most zwitterionic proteinogenic amino acids. For these HILIC or ion-pairing MS in positive ion mode would be useful complementary approaches.

**Metabolic changes in glioblastoma cells with *IDH1* mutations.** To demonstrate the utility of IC-MS in a discovery-driven untargeted context, we compared wild-type *IDH1* LN18 glioblastoma cells with those expressing the R132H mutant form of *IDH1*. Somatic mutations in genes encoding *IDH1* and *2* are found in over 80% of grade 2 and 3 glioma, some glioblastomas and over 13 other types of cancer[16–18]. The mutation is known for its gain-of-function effect leading to the production of 2-hydroxyglutarate (2-HG), now one of the best-known small-

molecule biomarkers in cancer. Despite this, exactly how 2-HG is linked to the mechanism of tumorigenesis and altered metabolic function is still unclear[19–21].

To investigate the links between the *IDH1* mutation and perturbed metabolism in more detail, we cultured six *IDH1* (R132H) mutant replicate dishes of glioblastoma cells and six *IDH1* wild-type LN18 glioblastoma replicate dishes of cells. Metabolite extracts from each dish were normalised and analysed using untargeted IC-MS. Univariate statistical analysis was performed on the resulting dataset as well as fold-change (FC) analysis for each compound feature. Out of the 1756 compound features measured, 40 were found to be significantly enriched and 146 were significantly depleted in *IDH1* mutant cells (FC > 2 and false discovery rate (FDR)-corrected *p* value < 0.05). 2-HG was strongly elevated in *IDH1* mutant cells, while 2-oxoglutarate was depleted (see box plots in Fig. 1e). In total, we identified 176 metabolites based on our database of authentic standards (we used the HMDB for the putative identification of a small number of additional metabolites). This analysis provided an overview of which identified metabolites were altered in abundance in the *IDH1* mutant cells, and Fig. 2 maps a subset of these identified metabolites onto the central carbon metabolic network demonstrating extensive coverage and showing which metabolites were enriched, depleted or where abundances were not changed when *IDH1* mutant and wild-type cells were compared (based on Kyoto Encyclopedia of Genes and Genomes (KEGG) metabolic pathways). Supplementary Data 2 provides the full list of 176 identified metabolites in the cell lines. Supplementary Fig. 3a–c and Supplementary Fig. 4a–c provide selected EICs and box plots illustrating metabolites that were altered in abundance and those that were not (all samples were normalised to the total DNA level). Using a volcano plot to visualise FC and *p* value (Fig. 3a, b, 'Batch 1'), 19 significantly altered identified metabolites were selected (increased or decreased; normalised FC > 2 versus FDR-adjusted *p* value < 0.05). All were depleted in mutant cells, except for 2-HG, which was enriched (Fig. 3a); a full list of significantly altered identified metabolites and compound features are given in Supplementary Data 3. 2-HG was the most significantly altered metabolite in the entire dataset (identified and unidentified compound features; Fig. 1e).

Correlation analysis (Fig. 3c for identified metabolites only) and Supplementary Fig. 5a (all compound features) showed a number of identified metabolites/compound features strongly correlated (positively and negatively) with changes in 2-HG levels (Pearson's *R* correlation > |0.8|). Negatively correlated metabolites included β-citryl-glutamate, oxoadipic acid, *N*-acetylaspartylglutamate, xylitol and 2-oxoglutarate (ranked by correlation coefficient in Fig. 3c). The negative correlations were not the result of loss of these metabolites in *IDH1* mutant cells, but these metabolites were significantly depleted when 2-HG was enriched (see, for example, box plot for β-citryl-glutamate in Supplementary Fig. 3b). This could be due to enzyme inhibition by 2-HG or loss of enzyme expression or another secondary effect; further studies are needed to determine the mechanism. In order to show the metabolic findings are reproducible, we repeated the entire tissue culture experiment and IC-MS analysis on a number of occasions over 12 months. The analysis of a second batch of cell samples (nine tissue culture replicates of *IDH1* mutant and nine replicates of *IDH1* wild type known as 'Batch 2') revealed agreement with significantly altered metabolites in 'Batch 1'. Nine out of the 16 significantly altered identified metabolites measured in 'Batch 2' matched those in 'Batch 1' (Fig. 3d and Supplementary Data 4 for Batch 2 dataset) (normalised FC > 2 and FDR-adjusted *p* value < 0.05).

Although univariate statistical analysis can identify metabolites that are significantly different in abundance between

experimental groups, and consistent differences were observed between replicate experiments several months apart, all univariate statistical approaches ignore correlations between metabolites within datasets. To investigate these, and create a model of differences between *IDH1* mutant and *IDH1* wild-type cells, we performed multivariate statistical analysis using partial least-squares discriminant analysis (PLS-DA), utilising all compound features (Batch 2 9 × 9 dataset). The PLS-DA scores plot and variable importance in the projection (VIP) ranking of compound features (Fig. 3e and Supplementary Fig. 5b, respectively), derived from this analysis, showed significant differences between mutant and wild-type cells. Analysis of a single metabolite (2-HG) was capable of distinguishing *IDH1* mutant from *IDH1* wild-type cells. Supplementary Fig. 5c shows results of the cross-validation test ($R^2 = 0.96$, $Q^2 = 0.94$, accuracy = 1.0) used to check the model was not over-fitting the data. Interestingly, when we removed 2-HG from the dataset, and remodelled the data, a minimum of only eight new compound features were required to differentiate *IDH1* mutant and *IDH1* wild-type cells (Fig. 4a shows the VIP plot). The model strength and accuracy was still high ($R^2 = 0.99$, $Q^2 = 0.96$ and accuracy > 0.99), using the same 'leave-one-out' cross-validation approach (Supplementary Fig. 5d). Note that phosphate ions (box plot Fig. 4b) and phosphate ion clusters, initially present in the highly ranked features (all with the same retention time 15.28 min), were also removed due to their very high abundance (likely derived from cell washing buffer), which influenced the model. The highest-ranked VIP values were for the eight identified metabolites/compound features: lactate, glutathione, NADH, uridine diphosphate, dGDP and β-citryl-glutamate and apparently two unidentified 'compound features' (14.33_146.0453 and 16.38_146.0454) (Fig. 4b shows box plots for each metabolite/compound feature). Of these, β-citryl-glutamate and NADH were significantly altered in *IDH1* mutant cells with an FDR-adjusted *p* value of $5.9 \times 10^{-10}$ and 0.0005, respectively).

An HMDB search at 5 p.p.m. mass accuracy provided a list of biologically relevant metabolites corresponding to *m/z* 146.0453 (Fig. 4c); given the relatively high *m/z* abundance, glutamate was the most likely identification and this was confirmed by retention time and accurate mass (<5 p.p.m.) matching to the analysis of an authentic standard (Supplementary Fig. 6a). The identification of glutamate was surprising due to the zwitterionic nature of glutamate (although highly acidic), so we further investigated additional proteinogenic amino acid standards. The majority did not provide a peak in the IC-MS spectrum, but four that did were aspartate, serine, lysine and glutamate (Supplementary Fig. 6a–d provides EICs). Unlike serine and alanine, the chromatographic peaks for glutamate (and to a lesser extent aspartate) were broad and split in both the standard and samples. For glutamate, the split peaks corresponded to the two retention times (14.33_146.0453 and 16.38_146.0454) identified as significant in the top 8 VIP scores (Fig. 4a). Although they were identified as significant, the poor peak shapes were treated with caution (possibly a result of suppressor effects) and no biological interpretations for glutamate or aspartate are made here. However, this does not rule out their importance in IDH mutant metabolism. By contrast with the broadness and poor peak shape of glutamate and aspartate, serine and lysine manifested well-characterised IC-MS peaks using standards and cell extracts (Supplementary Fig. 6c, d). This was an unexpected and useful finding for future studies.

The selected metabolites from PLS-DA modelling overlapped with the metabolites identified from the univariate statistical analysis (β-citryl-glutamate, *N*-acetylaspartylglutamate, NADH, ribulose-5-phosphate, 2-oxoglutarate and 2-HG), but it should be noted that multivariate modelling is influenced, in part, by the

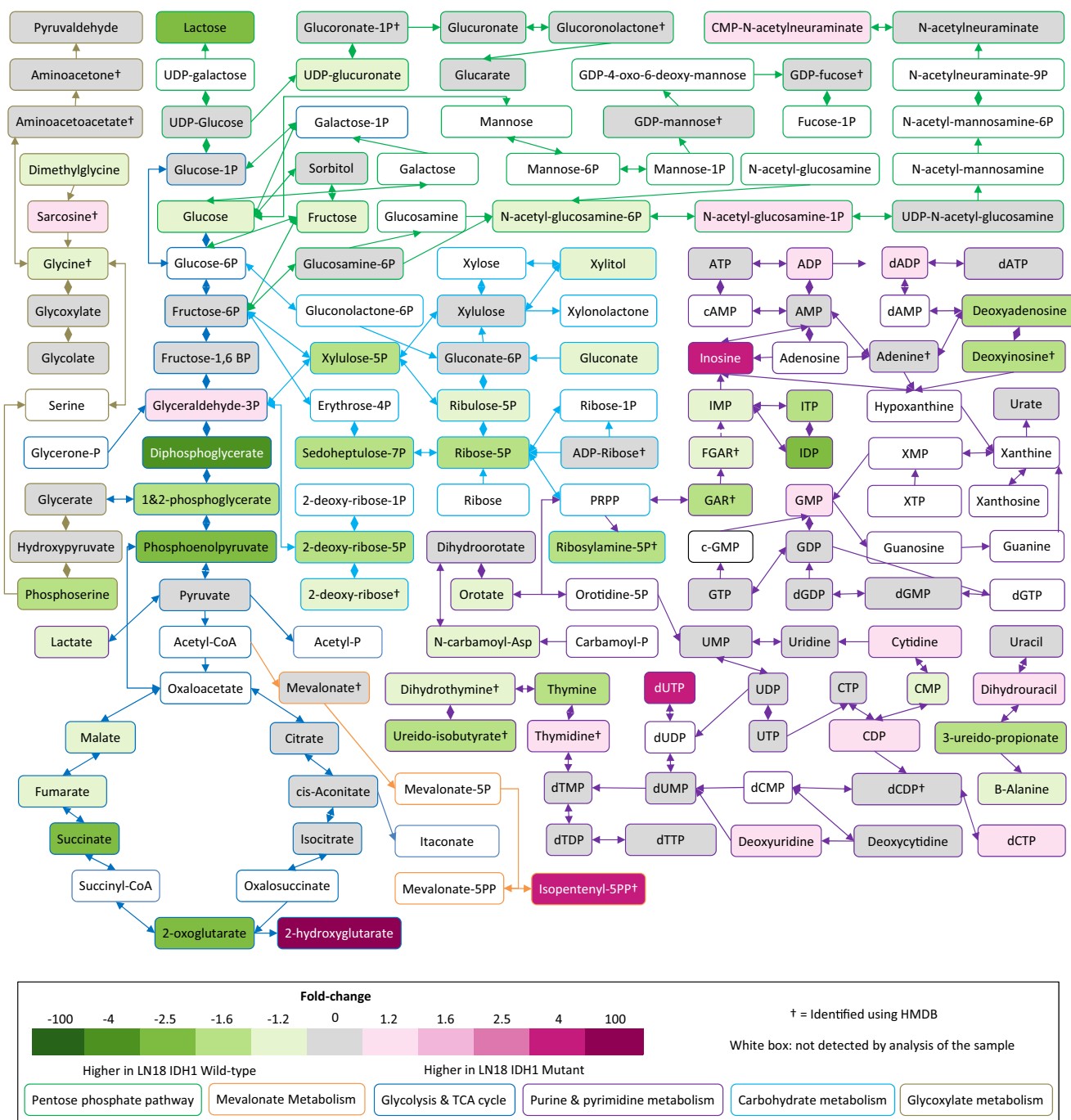

**Fig. 2 Fold-change heat-map representing central metabolic pathways and comparing LN18 *IDH1* mutant vs. *IDH1* wild-type cells.** Fold-changes >1.2 are in shades of pink and green for accumulation and depletion, respectively, with grey representing fold-changes <1.2 († indicates no authentic standard was available, but the metabolite was putatively identified by matching accurate mass (<5 p.p.m.), isotope pattern (>90%) and fragmentation (where possible) pattern matching (multiple HCD fragments matched to in-silico HMDB peaks) to the Human Metabolome Database (HMDB)). A white background denotes a metabolite not detected in the analysis.

abundance of the metabolite, although this is not an important factor associated with FC and *p* value statistics derived from univariate analysis. Results from univariate and multivariate statistical analysis were therefore not expected to necessarily overlap.

In order to better understand the possible functional importance of the changes in metabolite abundance observed, we performed metabolic pathways analysis. Prediction of metabolic pathway activity can be crudely investigated by simply mapping metabolite changes onto known metabolic pathways

(based on KEGG pathways for similar, for example, as in Fig. 2). However, this approach does not show the relative importance of metabolic changes associated with a particular pathway, as many metabolites are found in multiple pathways. As an alternative, we performed metabolic pathway analysis using organism specificity (*Homo sapiens*), enrichment and topographical analysis; the latter are algorithms that aim to represent the importance of a particular pathway based on its metabolic connectivity. Figure 5 provides a visual representation of the results of this analysis displaying metabolic pathways that were significantly altered,

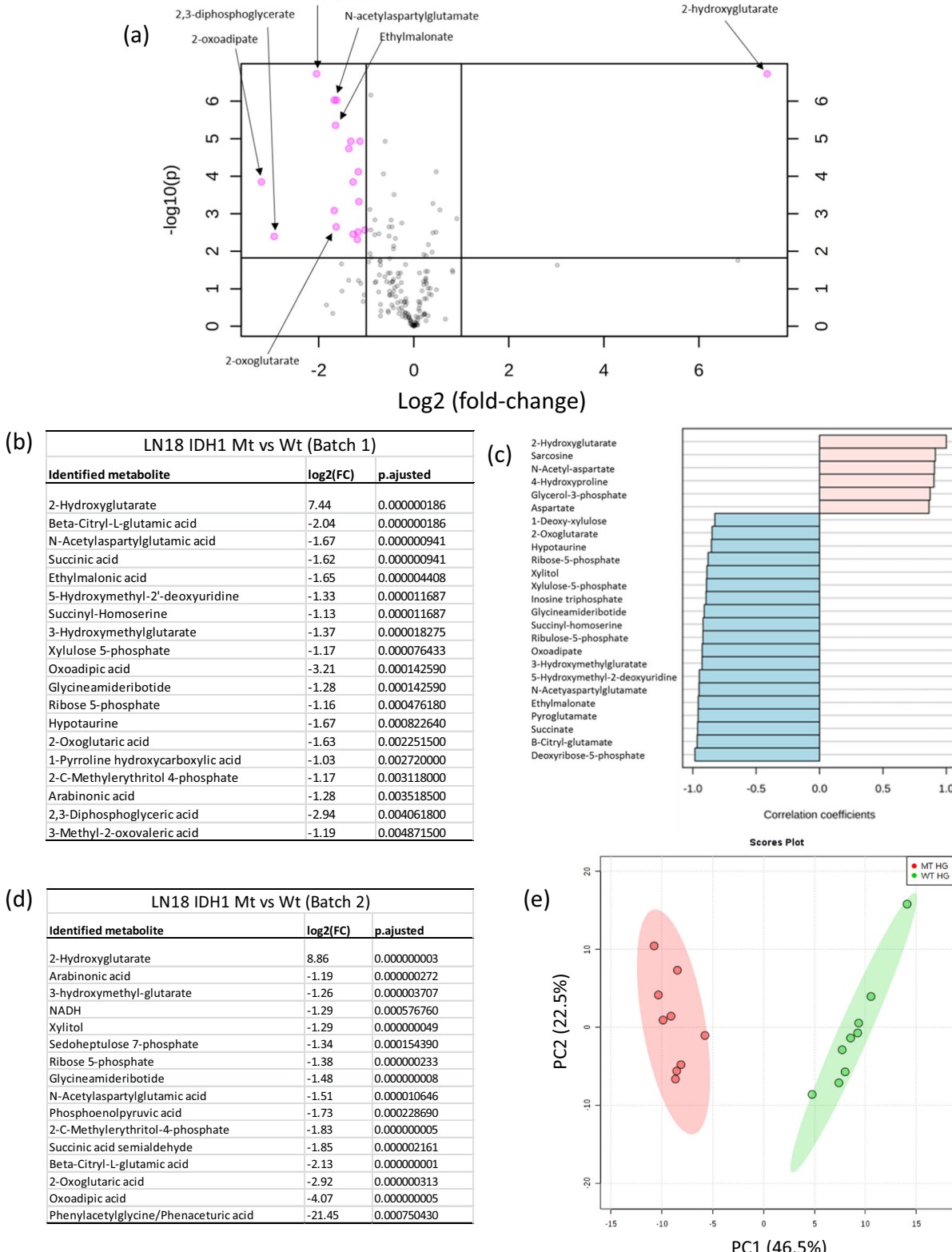

**Fig. 3 Identification of altered metabolite abundances and pathways analysis in IDH mutant glioma cells. a** Volcano plot showing identified metabolites comparing log 2 FC and −log 10 FDR-corrected *p* values (*t* test, *p* < 0.05) from normalised data. Parametric and non-parametric tests were investigated to determine significance; little change in significance was observed between the methods for the identified compounds reported here. By combining fold change and significance, the volcano plot enables only compounds that are significantly altenated in abundance to be identified. Fifteen identified metabolites were identified as being significantly altered in *IDH1* mutant cells. **b** List of most significantly altered identified metabolites in *IDH1* mutant cells for Batch 1 (FC > 2 and FDR-adjusted *p* value < 0.05). **c** Correlation analysis shows that most identified metabolites and majority of all compound features are depleted in *IDH1* mutant cells compared to *IDH1* wild type. **d** List of most significantly altered identified metabolites in *IDH1* mutant cells for Batch 1 (FC > 2 and FDR-adjusted *p* value < 0.05). **e** PCA plot comparing all compound features from the untargeted IC-MS/MS analysis of LN18 IDH mutant and wild-type cells ('Batch 2').

towards the top of the figure, and those that have a higher metabolic impact towards the right-hand side of the figure (based on KEGG metabolic pathway ontology using a MetaboAnalyst workflow). The results indicate that the most important metabolic pathways affected in the *IDH1* mutant cells compared to wild-type cells are tryptophan metabolism, lysine degradation, butanoate metabolism, PPP metabolism and, to a lesser extent, TCA cycle and glyoxylate metabolism.

Supplementary Figs. 7 and 8 show the selected annotated pathways, colour coding the measured metabolites and their significance, and Supplementary Data 5 provides a table with the significance values associated with the pathway analysis. Upon closer inspection these metabolic pathways can be split into two groups. The first pathways are associated with central carbon metabolism and include glyoxylate, butanoate, aldarate, PPP, and TCA cycle metabolism. The second pathways are associated with amino acid metabolism (lysine degradation and tryptophan metabolism).

The metabolites significantly altered in the central carbon metabolic pathways (group 1) partially overlap in terms of represented metabolites (Supplementary Figs. 7 and 8); they are most comprehensively represented (in terms of coverage) by the PPP and TCA cycle. The PPP is a cytoplasmic, mostly anabolic pathway that generates NADPH and ribose-5-phosphate for nucleotide synthesis. It is therefore potentially linked to mutant *IDH1* 2-HG production as NADPH is consumed in large quantities by the mutant IDH enzyme which forms 2-HG[21]. Changes in TCA cycle metabolite abundances appear to affect the left hand side of the pathway from 2-oxoglutarate to malate where the metabolites are depleted in *IDH1* mutant cells. This is not unexpected as *IDH1* has been shown to be expressed in the cytoplasm in preference to mitochondria (unlike *IDH2*)[19]. The second group of metabolic pathways was associated with amino acid metabolism, most notably tryptophan and lysine degradation pathways, and most significantly, by the single metabolite 2-oxoadipate, which is an intermediate in both pathways and well characterised by IC-MS.

## Discussion
In this study, we developed, evaluated and applied anion-chromatography coupled to high-resolution tandem MS for untargeted metabolomics. Validation of the method demonstrates robustness, stability and comprehensive coverage of the central human carbon metabolic network. We demonstrated that the method is useful for the analysis of cells, tissues and bio-fluids. Characterisation of 431 metabolite standards was demonstrated, significantly extending metabolite coverage associated with IC-MS analysis. To demonstrate application of the method for untargeted metabolomics we analysed *IDH1* mutant and *IDH1* wild-type LN18 glioblastoma cells.

The analysis of LN18 *IDH1* mutant (R132H) and wild-type glioma cells demonstrated comprehensive coverage of primary metabolic pathways and revealed alterations in PPP, TCA cycle, glyoxylate and aldarate metabolism. Changes in a range of metabolites were identified, including significantly elevated 2-HG, depleted β-citryl-glutamate, oxoadipate and 2-oxoglutarate. Multivariate modelling was used to build a validated model that could distinguish *IDH1* mutant from *IDH1* wild-type cells using 2-HG alone. The results support the proposal that 2-HG is a useful biomarker of IDH mutations[19]. Interestingly, it was also found that by removing 2-HG from the dataset and re-modelling the data a new 8-component model could differentiate *IDH1* mutant from wild-type cells; the identified metabolites in the model used were lactate, glutathione, NADH, uridine diphosphate, dGDP, β-citryl-glutamate, lactate and glutamate. Lactate

depletion has been observed in previous studies where IDH mutant cells were interpreted as less 'glycolytic' (compared to wild-type cells) and therefore altered, from the typical phenotype associated with the 'Warburg effect', towards a greater reliance on TCA cycle function[22,23].

Although we showed that zwitterionic proteinogenic amino acids were generally not amenable to IC-MS (with the exception of glutamate, aspartate, alanine and serine; Supplementary Fig. 6), the levels of selected amino acid degradation products (and other modified forms of amino acids such as *N*-acetylated amino acids) were well characterised; some were shown to be significantly depleted in *IDH1* mutant cells. These included 2-oxadipate, β-citryl-glutamate, *N*-acetylaspartylglutamate, glycineamideribotide, succinyl homoserine, phenylacetylglycine and 3-hydroxymethylglutarate. Our observed depletion of *N*-acetylated amino acids is concordant with the finding of a previous study by Reitman et al.[24], which reported that *N*-acetylated amino acids are significantly depleted in *IDH1* mutant cells compared to controls (human oligodendroglioma cells). However, we cannot find a literature precedent for changes in 2-oxoadipate, β-citryl-glutamate, glycineamideribotide, succinyl homoserine and phenylacetylglycine levels being associated with IDH mutations to date. β-Citryl-glutamate showed particularly robust depletion in *IDH1* mutant cells being identified by both univariate and multivariate statistical analysis as significant. Taken as a whole, there is a predominance of modified amino acid-related metabolites in the list of significantly altered identified metabolites in *IDH1* mutant cells. It is interesting to note that elevated 2-HG in IDH mutant glioma has been shown to inhibit branched-chain-amino-acid aminotransferase 1 and 2 (BCAT 1 and 2), which are transaminases that convert branched-chain amino acids (leucine, isoleucine and valine) to their keto-acids in the cytoplasm and mitochondria, respectively[25].

Pathways analysis using all identified metabolites revealed altered tryptophan and lysine degradation pathways as highly significantly changed in *IDH1* mutant cells compared to wild-type cells. On closer inspection, this appears to be due to 2-oxoadipate (found in both pathways) being highly depleted in *IDH1* mutant cells, suggesting an *IDH1* mutant cell-specific effect. In the lysine degradation pathway, 2-oxoadipate is formed from 2-aminoadipate via the enzyme 2-aminoadipate transaminase and interestingly this transaminase requires 2-oxoglutarate as a co-factor[26]. It is therefore possible that the depletion in 2-oxoglutarate levels observed in *IDH1* mutant cells may subsequently affect formation of 2-oxoadipate by limiting 2-aminoadipate transaminase activity. It is also possible that, analogous to the suggested mechanism for BCAT 1 transaminase inhibition, the structural similarity between 2-oxoglutarate and 2-HG leads to elevated 2-HG levels inhibiting 2-aminoadipate transaminase. To our knowledge, it has not been previously shown that 2-oxoadipate levels are significantly depleted in IDH mutant cells or whether 2-oxoadipate production is inhibited by increased 2-HG. Given that 2-oxoglutarate and 2-oxoadipate are closely related by structure, and that elevated 2-HG has been proposed to promote tumorigenesis via inhibition of 2-oxoglutarate-dependent oxygenases involved in transcriptional/chromatin regulation[21], it may be of interest in future studies to investigate the potential role for 2-oxoadipate in modulating oxygenase activity.

In conclusion, we have developed an IC-MS method that enabled robust and comprehensive, untargeted and discovery-driven, metabolomics analysis. We demonstrated its use to effectively and comprehensively map the metabolic consequences of a specific cancer-related metabolic gene mutation (*IDH1*). More generally, given the predominance of negatively charged ionic metabolites in primary carbon metabolism, and the increasing recognition that perturbations in energy

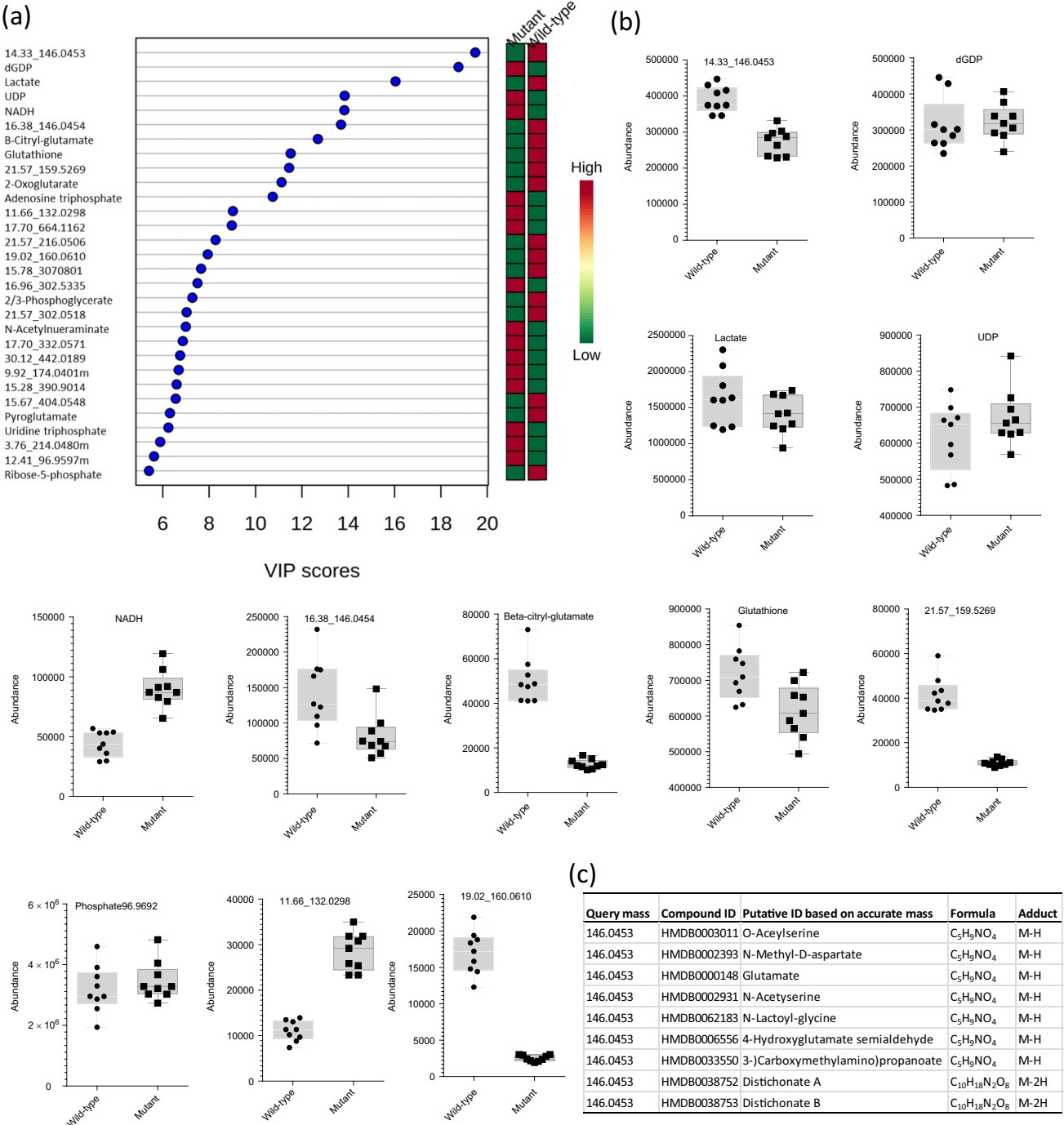

**Fig. 4 Supervised multivariate analysis (PLS-DA) reveals significantly altered metabolites in *IDH1* mutant cells. a** PLS-DA variable importance in the projection (VIP) plot showing the top 30 compound features and identified metabolites without 2-HG. The results reveal the importance of lactate, glutathione, NADH, uridine diphosphate, dGDP and β-citryl-glutamate and two unidentified compound features (14.33_146.0453 and 16.38_146.0454) that differentiate *IDH1* mutant from wild-type cells. Coloured boxes to right-hand side of each metabolite qualitatively indicate degree and direction of change in abundance between the two experimental groups. **b** Box plots for compound features and identified metabolites important in differentiating IDH mutant and wild-type cells in the PLS-DA model (based on VIP score). **c** List of putative identifications for the metabolite feature '14.33_146.0453' based on 5 p.p.m. accurate mass search of the HMDB database[32].

transduction and associated pathways are prevalent in a wide range of diseases including cancer, we suggest IC-MS has important potential as a platform for untargeted metabolomics. Areas of application that may benefit include: investigating disease aetiology associated with central carbon metabolism, monitoring therapeutic interventions, the study of inborn errors of metabolism and analysis of the gut microbiome. These

all involve altered ionic metabolism, the analysis of which is well suited to IC-MS.

## Methods
**Materials and standards**. Metabolite standards, cell culture media and reagents, including Dulbecco's modified Eagle's medium (DMEM) and foetal bovine serum (FBS), as well as high-performance liquid chromatography (HPLC)-grade solvents

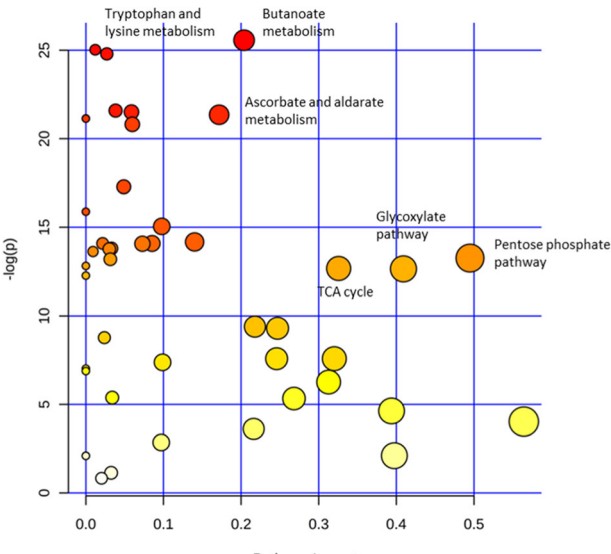

**Fig. 5 Pathway analysis using all identified metabolites showing metabolic pathways represented as nodes.** The size of the circular nodes represents the proposed pathway impact (based on pathway topological analysis). The colour shows the significance (based on pathway enrichment analysis[33]). Tryptophan, lysine, butanoate, pentose phosphate pathway and glyoxylate metabolism are all identified as significantly altered in IDH mutant cells.

(methanol and acetonitrile, formic acid), were purchased from Sigma-Aldrich (Gillingham, Dorset, UK). Metabolite standards were made up to a concentration of 1 μg/mL in 80% HPLC MeOH and equal volumes of 8 standards were mixed together in a single vial for analysis.

**Metabolite standards database.** We constructed a metabolite standards database using Microsoft Excel, which included, for each metabolite, values for mono-isotopic mass (calculated from the chemical formula), isotope pattern (calculated from the chemical formula), chromatographic retention time (experimentally determined from analysis of the authentic standard) and HCD fragmentation pattern (experimentally determined from analysis of the standard). New untargeted datasets were searched using this database of values for compound feature matches.

**Metabolite identification.** Metabolite identifications were accepted when the following criteria were met: <5 p.p.m. differences between measured and theoretical mass (based on chemical formula), <30 s differences between authentic standard and analyte retention times, isotope peak abundance measurements for analytes were >85% matched to the theoretical value generated from the chemical formula. When measured, fragmentation patterns were matched to the base peak and at least two additional peaks in the MS/MS spectrum to within 12 p.p.m. (note: the top 10 data-directed fragmentation method was not able to provide fragment ions for all ions measured in the MS 1 spectrum).

**IDH1 R312H and IDH1 wild-type LN18 cells.** Immortalised LN18 (glioblastoma cell line) were obtained from ATCC (CRL-2610) and genetically modified to overexpress mutant IDH1 by lentiviral vector transduction. Complementary DNA encoding to the R132H mutant sequence of IDH1 were cloned in the lentiviral transfer vectors pCC.sin.36.MCS.PPTWpre.CMV.tTA-S2tet following the methods reported in Bardella et al.[27]. Vectors' stock and titration were prepared as described in Bardella et al.[27]. The IDH1 wild-type cells used for comparison were the untransduced LN18 glioblastoma cell line. All cells were grown in DMEM supplemented with 5% of FBS and 1% GlutaMAX. Cells were grown to 90% confluency.

**Metabolite extraction.** Metabolism was arrested in tissue culture dishes at the point of harvesting by pouring off the media, and by rapid washing with ice-cold phosphate-buffered saline buffer. Liquid nitrogen was then added to the adherent cell layer to cover all cells. As the liquid nitrogen was evaporating 500 μL of ice-cold 80% (v/v) aqueous MeOH was then added. The cells were scraped thoroughly in the cold MeOH solution to aid lysing of cells and their removal, making sure >95% of the cells were removed from the surface of the dish. The resulting MeOH cell suspension was then pipetted into a microfuge tube and centrifuged at $21,693 \times g$ for 25 min at 4 °C. The supernatant was removed and the concentration of DNA was measured at 260 nm using a Nanodrop (Thermo Fisher Scientific, MA, USA). Each sample was then normalised to the sample with the lowest total DNA content by dilution with MeOH. Soluble proteins were removed prior to MS analysis using

10 kDa MwCO microfuge filters (Amicon, Merck). A QC sample was created by combining 5 μL of each sample in a single autosampler vial. This was injected at the start of the LC-MS/MS sequence.

**IC-MS method.** A 5 μL partial loop injection was used for all analyses. Chromatographic separations were performed using a Dionex ICS-5000+ Capillary HPIC system (Dionex, Sunnyvale, CA, USA) coupled to a Q-Exactive HF hybrid quadrupole-Orbitrap mass spectrometer (Thermo Scientific, San Jose, CA, USA). A Dionex IonPac AS11-HC column ($2 \times 250$ mm$^2$, 4 μm; Dionex, Sunnyvale, CA, USA) at 30 °C was used with an aqueous hydroxide ion gradient at a flow rate of 0.25 mL/min with the following steps: 0 min, 0 mM; 1 min, 0 mM; 15 min, 60 mM; 25 min, 100 mM; 30 min, 100 mM; 30.1 min, 0 mM; 37 min, 0 mM. A continuously regenerated trap column was used to remove ionic contaminants from the eluent and ion suppression was achieved using a Dionex ERS 500e (Dionex, Sunnyvale, CA, USA) in external water mode with a flow rate of 0.5 ml/min. The mass spectrometer was equipped with a HESI II probe in negative ion mode with source parameters set as follows: sheath gas flow rate, 60; auxiliary gas flow rate, 20; sweep gas flow rate, 0; spray voltage, 3.6 kV; capillary temperature, 300 °C; S-lens RF level, 70 and heater temperature 350 °C. MS scan parameters were set as follows: microscans, 2; resolution, $7 \times 10^4$; AGC target, $1 \times 10^6$ ions; maximum IT, 120 ms and scan range, 60–900 m/z. MS/MS scan parameters were set as follows: microscans, 2; resolution, $1.75 \times 10^4$; AGC target, $1 \times 10^5$ ions; maximum IT, 250 ms; loop count, 10; MSX count, 1; isolation window, 2.0 m/z; collision energy, 35; minimum AGC target, $5 \times 10^3$ ions; apex trigger 1–15 s; charge exclusion, 3–8, >8 and dynamic exclusion, 20.0 s.

**HILIC-MS method.** A 5 μL partial loop injection was used for all analyses. Chromatographic separations were performed using a Dionex Ultimate 3000 UHPLC system (Dionex, Sunnyvale, CA, USA) coupled to a Q-Exactive HF hybrid quadrupole-Orbitrap mass spectrometer (Thermo Scientific, San Jose, CA, USA). A BEH Amide column ($2.1 \times 100$ mm$^2$, 1.7 μm; Waters, Milford, MA, USA) at 25 °C was used with mobile phase A: 95% acetonitrile (v/v) aqueous containing 10 mM ammonium acetate and mobile phase B: 50% acetonitrile (v/v) aqueous containing 10 mM ammonium acetate. The linear gradient used was: 0 min, 1% B; 1.0 min, 1% B; 6.0 min 45% B; 10.0 min, 95% B; 12.0 min, 99% B; 12.1 min, 1% B; 15.0 min, 1% B. The flow rate was 0.4 mL/min and the total run time was 15 min. The mass spectrometer was equipped with a HESI II probe operating in negative and positive ion modes with source parameters set as follows: sheath gas flow rate, 25; auxiliary gas flow rate, 8; sweep gas flow rate, 0; spray voltage, ±3.5 kV; capillary temperature, 300 °C; S-lens RF level, 55 and heater temperature 300 °C. MS scan parameters were set as follows: microscans, 2; resolution, $7 \times 10^4$; AGC target, $1 \times 10^6$ ions; maximum IT, 120 ms and scan range, 60–900 m/z. MS/MS scan parameters were set as follows: microscans, 2; resolution, $1.75 \times 10^4$; AGC target, $1 \times 10^5$ ions; maximum IT, 80 ms; loop count, 10; MSX count, 1; isolation window, 2.0 m/z; collision energy, 35; minimum AGC target, $5 \times 10^3$ ions; charge exclusion, 3–8, >8 and dynamic exclusion, 20.0 s.

**C18-MS/MS method.** A 5 μL partial loop injection was used for all analyses. Chromatographic separation was performed using a Dionex Ultimate 3000 UHPLC system (Dionex, Sunnyvale, CA, USA) coupled to a Q-Exactive HF hybrid quadrupole-Orbitrap mass spectrometer (Thermo Scientific, San Jose, CA, USA). A CORTECS T3 C18 column ($2.1 \times 100$ mm$^2$, 1.6 μm; Waters, Milford, MA, USA) at 40 °C was used with mobile phase A: water with 0.1% (v/v) aqueous formic acid and mobile phase B: methanol with 0.1% (v/v) aqueous formic acid. The linear gradient used was: 0 min, 5% B; 4.0 min, 50% B; 12.0 min 99.9% B; 14.0 min, 99% B; 15.1 min, 5% B. The flow rate was 0.3 mL/min and the total run time was 18 min. The mass spectrometer was equipped with a HESI II probe in negative ion mode with source parameters set as follows: sheath gas flow rate, 25; auxiliary gas flow rate, 8; sweep gas flow rate, 0; spray voltage, 3.5 kV; capillary temperature, 300 °C; S-lens RF level, 70 and heater temperature 300 °C. MS scan parameters were set as follows: microscans, 2; resolution, $7 \times 10^4$; AGC target, $5 \times 10^6$ ions; maximum IT, 120 ms and scan range, 60–900 m/z. MS/MS scan parameters were set as follows: microscans, 2; resolution, $1.75 \times 10^4$; AGC target, $1 \times 10^5$ ions; maximum IT, 80 ms; loop count, 10; MSX count, 1; isolation window, 2.0 m/z; collision energy, 35; minimum AGC target, $5 \times 10^3$ ions; charge exclusion, 3–8, >8 and dynamic exclusion, 20.0 s.

**Data processing.** Raw data files were processed using Progenesis QI (Waters, Elstree, UK). This involved alignment of retention times, peak picking including identification of the presence of natural abundance isotope peaks, characterising adducts and identification of metabolites using our in-house database. Data were further processed as follows: missing values were eliminated using a Progenesis QI co-detection automated method. Samples were statistically normalised to all ions using Progenesis QI. Data were not transformed and was scaled using 'Pareto scaling' in MetaboAnalyst.

**Statistics and reproducibility.** Univariate statistical analysis included determining FC and t tests between experimental groups for compound features and combined in volcano plots. An FC threshold of 2 and an FDR-adjusted p value cut-off of 0.05 were

used to determine significance. Unsupervised multivariate statistical analysis included PCA, which was used to visualise global metabolic profiles and determine sample outliers. PCA plots were evaluated alongside heat-maps comparing all compound features and samples to assess the impact of sample normalisation. Supervised multivariate statistical analysis using PLS-DA was used to model important compound feature that differentiated *IDH1* mutant and wild-type cells. Feature loadings and VIP scores were ranked to determine the most important features. Models were assessed based on cross-validation with $R^2$ and $Q^2$ and Accuracy reported. Data and sample correlation were visualised using heat-maps with a Euclidean distance measure and all compound features.

**Pathway analysis**. MetaboAnalyst was used for pathway topology analysis[28,29]. This was based on the KEGG metabolite library specific to *Homo sapiens* and used Globaltest enrichment analysis[30]. Intensity data for all identified metabolites was normalised by sum and Pareto scaled. In addition to allocating metabolites to metabolic pathways, an estimate of the importance of a metabolite to the metabolic network was estimated using a relative betweenness centrality algorithm[31].

**Reporting summary**. Further information on research design is available in the Nature Research Reporting Summary linked to this article.

## Data availability
The summary statistics and metadata generated from the untargeted mass spectrometry experiments reported in this study (referred to as Batch 1 and Batch 2) are available in Supplementary Data files 1–5 and the Supplementary Figures and Tables. The raw mass spectrometry data files from which the results were generated have been deposited in *Metabolites* (Reference: MTBLS1654), a freely accessible public data archive.

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

## Acknowledgements
We thank Joe Harvey, Ian Tomlinson and Olaf Ansorge. This work was supported by grants from UK BBSRC (BB/R013829/1) (J.S.O.M.), The Wellcome Trust (106244/Z/14/Z and 204483/Z/16/Z) (C.J.S. and J.S.O.M), Cancer Research UK (C8717/A18245) (C.J.S.) and the John Fell Fund (JFF 142/116) (J.S.O.M.).

## Author contributions
J.S.O.M., C.J.S, T.C.-H. and J.W.-T. designed and planned the experiments. J.W.-T., J.G. and E.P. developed and conducted the metabolomics method and experiments. J.W.-T., I.H., C.B. and M.I.A. created mutant cell lines and prepared cell cultures for analysis. J.W.-T., T.C.-H., D.H. and A.N. performed data processing and analysis. J.S.O.M wrote the manuscript and all the authors edited and approved the final version.

## Competing interests
The authors declare no competing interests.
