## [Peer Review File · Communications Biology]

Reviewers' comments:

Reviewer #1 (Remarks to the Author):

The manuscript entitled „Untargeted metabolomics by anion-exchange chromatography-mass spectrometry provides metabolic insights into gene mutations in cancer“ presents a very interesting untargeted method for comprehensive and robust analysis of primary metabolic pathways. The developed methodology was deeply validated on different types of the samples including cells, tissues and bio-fluids. I do believe that this work is in high

The manuscript is nicely structured, and results are described in a logical manner. Such a large amount of results is represented in a very concise and clear way. The method was nicely described and profoundly validated and represented results only confirm its robustness.

I have no objections to the substantive part of the article: introduction is properly structured and provides bases for the rest of the manuscript; materials and methods together with supplementary information provide a lot of information (although some details are missing – see comments below); results clearly illustrate the method development, validation and application process. I do believe this manuscript is high importance and interest for the readers of Communication Biology and I recommend its publication after major revision:

Manuscript

- Title: In my opinion, the title of the manuscript does not fit to the content of it: From the main text and the reasoning presented in the text, I understood that the work is dedicated to the development and validation of the method for the measurement of the metabolites of primary metabolic pathways. The application of this methodology to investigate the metabolic effects of isocitrate dehydrogenase 1 (IDH1) mutations in glioblastoma cells was performed as a proof of the concept. Therefore, I would change the title, because now the title is focused only on the last part of work.

- Line 6: Please move affiliation number 2 to the next line

- Line 46: repetition of the word “challenge”

- Please comment also work of Fu and colleagues on the method for Targeted Determination of Tissue Energy Status by LC-MS/MS (<https://doi.org/10.1021/acs.analchem.9b00217>)

- Lines 96-100: „We used a range of metabolites with differing physicochemical properties and cell extracts for the determination of sensitivity, linearity, retention time reproducibility (robustness) and lower limit of detection and quantification (LLOD/LLOQ). Results from the targeted validation (Supplementary Fig.1) include retention time reproducibility, linearity and the lower limit of detection (Supplementary Fig. 1b, 1c and 1d).“ You determined sensitivity, linearity, retention time reproducibility (robustness) and lower limit of detection and quantification (LLOD/LLOQ), but results include only retention time reproducibility, linearity and the lower limit of detection. Why? Please comment on this and provide the missing information.

- Figure 1: Please either split the figure into two or three smaller one or increase the size of the font because the graphics and axis are not readable because even using zoom it is impossible to read the legend for the PCA plots to read the labelling for the sample groups.

- The caption to the figure 1 Lines 168-169: Please change the current description of HILIC and RP methods to C18-MS/MS and HILIC-MS/MS because now information about tandem analysis is missing

- Please unify the nomenclature for hexose monophosphates to avoid the presence of different representations such as e.g. glucose-1P and mannose-1-P.

- Line 206: For comparison of such small sample sets parametric test should not be used, even if the normality was tested. Non-parametric methods (e.g. Mann-Whitney U-test) should be used instead. In this particular case, observed differences are very strong and I do believe they will be captured also with U-test, however in case of weaker models the difference between parametric and non-parametric test might be significant.

- Line 225: Please add to the correlation coefficient the symbol of absolute value $>|0.9|$

- Lines 244 and 248: Why two different validation methods were used? Can you justify the application

of the permutation test on such a small group of samples?

- Line 255: remove spare . in the Fig.. 4c
- The caption to figure 2 Lines 311-312: "A white background denotes a metabolite not identified in the analysis." Do you mean identified as assigned to the true metabolites or identified as measured in the samples (metabolites not detected)? This point is not clear.
- Figure 3: panel A-labels of the discriminating compounds are not visible – please increase the font. The same I can comment on the names of the metabolites in panel b and d. Please make adequate changes.
- The caption to the figure 3 Line 315: Please add – to the log10 (it should be -log10...)
- The caption to figure 3 Line 319: You stated that the hierarchical clustering was performed for the top 50 identified metabolites, why the figure shows the map for only 30 metabolites. Can you explain why?
- In some parts of the manuscript (e.g. Figure 1, panel A) is stated that the RP method was used in the UPLC mode. Did you indeed performed UPLC analysis and extended 600 bars? If not, please make adequate corrections.
- VIP score on its own cannot be used to select statistically significant variables. It must be combined with some other measures such as p-value from univariate analysis or correlation coefficient from multivariate analysis. Please check if still all selected variables can be considered as statistically significant after combining VIP with other values.
- Line 468: Please change rpm value to x g since rpm gives very different centrifugal force depends on the size of the rotor.
- Paragraph Untargeted metabolomics method development (lines 476-486): I do not like how this part is described. Within the entire manuscript, a lot of information is provided but here very little details are given. If you decided to include in the manuscript the deplanement of the method you should provide adequate details, both about what was done and what was exact results. Stating that "for example, a MeOH precipitation was not sufficient...." (line 480) is not enough. Please, either provide information about method development, giving details about tested values etc, or remove this part from the manuscript.
- To avoid repetitions I suggest adding in the Method, a section called e.g. Instrumentation and move to it information about HPLC and MS system and stated that all the analysis were performed on..... In this way, the same information does not have to be repeated three times.
- In the description of MS parameters (in all three chromatographic methods) units are missing - line 499, 516 and 533). Please make adequate corrections.
- Figure 1C and Figure 3D are not referred in the text – Please add adequate comments in the text or remove mentioned panel from the manuscript.
- Figure 3: Change the order of panel on the figure and their labelling to quote them in the text in order of reading. No panel A and C are quoted as first (line 218), then Panel B and (223) and finally E (line 280). As mentioned above, panel D is not quoted.

Supplementary information

- Figure 1 Panel A – The figure presented in panel A does not match the description in the caption to this figure. In the manuscript, this figure is referred (Line 90) as a comparison of filtered and non-filtered samples, while the caption refers to the RT reproducibility. Please clarify this and make adequate changes.
- The caption to Figure 1 Panel B – I do believe that this text corresponds to the panel E on the figure. Please correct this.
- Figure 1 Panel B – please increase the size of the font for the names of the compound on the x-axis, Panel F – increase the size of the legend of the sample groups and axis.
- Figure 1 Panel D – Why the RT variability is given only for 6 out of 14 shown metabolites?
- Figure 1 Panel E – please add also information about the number of features with %CV below 15 and

30 represented as a % of the total number of detected features. And comment on these results.

- The caption to Figure 1 Panel D – Change Rt to RT.
- The caption to Figure 1 Panel G – The description of the table from the panel G is not clear. Please re-write it.
- Figure 3 Panel A: Figure shows EIC for 7 ions, but the last one (in black) is not labelled. The caption describes only 6 compounds. Can you explain this?
- Figure 3 Panel A and Caption to this panel: Metabolites on the figure are labelled as A, B, C, while in the caption as 3C1, 3C2..... Please unify the labels to avoid any misunderstanding.
- Figure 5 Panel B and F: Please increase the font because now these two panels are not readable.
- Figure 5 Panel D and Figure 6 Panel A: What is the meaning of colour bars on the right side of the plot? Do the colours correspond to the correlation or abundance? Please clarify this.
- Figure 8-10: The codes of the compounds should be represented as names of the metabolites. This will help to read and interpreted presented results.
- Figure 8 Panel C and Figure 10 Panel A: I do not see the reason to present pathways where single (Fig 8) or just two (fig 10) metabolites were matches.
- Figure 10: Please remove the label for panel A. It is redundant since this figure has only a single graphic.
- Panel A and F from figure 5 are not quoted in the text. Please make adequate changes.

- Check the representation of R2 and Q2 within the manuscript and supplementary information to ensure that all are represented as superscript: R² and Q²

Reviewer #2 (Remarks to the Author):

The manuscript by Walsby Tickle and colleagues describes the use of anion exchange chromatography in metabolomics. They report the measurement of 431 metabolites with good coverage of mammalian central carbon metabolism. They leverage top 10 data-directed MS² to confirm the identity of many of these metabolites and demonstrate advantages to HILIC chromatography, particularly retention time stability. The authors then apply this approach to characterize IDH1 mutations in LN18 glioblastoma cells observing increased 2-hydroxyglutarate and decreased alpha-ketoglutarate in the context of overexpression of mutant IDH1 R132H. Then, using an untargeted data analysis approach the authors also implicate changes in the lysine and tryptophan degradation pathways.

The manuscript is well-written, and the experiments are clearly described and appear in a logical sequence. The RP-IP and HILIC methods used as a comparison are appropriate and informative. Metabolites that are challenging to measure using this method or give bad peak shapes have been reported. The statistical analysis methods used are appropriate.

I have the following suggestions that could strengthen an excellent manuscript:

1. Since the manuscript is centered around the benefits of anion exchange chromatography, which has not been widely employed in metabolomics, it could benefit with a more explicit introduction to how this separation differs from other commonly used separations. The methods section is comprehensive, but I think it would be of general interest to the readers to have this also outlined in the introduction. Along the same lines:

Line 84: ion-suppressor would be clarified, the typical LC-MS metabolomics audience will be familiar

with the concept of ion suppression in ESI due to high abundance analytes and may be confused. Same for anion suppressor on line 86.

Line 182: clarify why the technique is not suitable for zwitterionic and positive compounds. A comment on how ion exchange chromatography for the analysis of cationic species would be helpful here.

2. The cell model needs to be defined more carefully:

Line 104: "Two different cell types (LN18 glioma cells genetically modified by lentiviral vectors transduction to overexpress the R132H mutant form of isocitrate dehydrogenase 1 (IDH1) and untransduced IDH1 wild-type cells)."

However, line 453: "Immortalized LN18 (glioblastoma cell line) were obtained from ATCC (CRL-2610) and genetically modified to overexpress either wild type or mutant IDH1 by lentiviral vector transduction. Briefly cDNAs encoding to the wild type or R132H mutant sequence of IDH1 were cloned in the lentiviral transfer vectors pCC.sin.36.MCS.PPTWpre.CMV.tTA-S2tet following the methods reported in Bardella et al. 2016."

The former indicates the control cell line is unmanipulated, the later indicates it had overexpression of WT IDH1. The experiments are valid in either case, but the mutant protein is clearly overexpressed (on top of endogenous wild type IDH1) and the distinction of whether this comparison is to endogenous levels or overexpression of the WT IDH1 is important and should be clear.

3. Stacked chromatogram plots could be improved. In supplementary figure 3a the black trace is missing a label, in all cases (Supplementary Fig 3a and 4a) labeling the plots with the metabolite names would help improve clarity.

4. The correlation plots (Fig 3b and supplementary Fig 5b) are dominated by correlations of +1.0 or -1.0 with 2-HG levels. This almost perfect correlation could benefit from a little more explanation – are the positively correlating metabolites in 5b in-source fragments / adducts of 2HG? Are the negatively correlating metabolites in both plots -1.0 because detection of those metabolites is lost in the mutant IDH condition?

5. Figures rely heavily on plots from MetaboAnalyst. While the data analysis methods chosen are appropriate, a little more explanation of what is being tested and displayed would be helpful to those who do not regularly use this on-line package, particularly in the figure legends.

Minor:

Why was the resolution setting for the QE HF so much lower (17,500) for IC but 70,000 for RP and IP methods?

The authors could consider replacing IC-MS/MS with IC-MS analysis (and the same for HILIC and RP methods, as it appears on lines 52/104. In my opinion the use of MS/MS in the abstract suggests a triple quad analysis, but since the authors are trying to highlight broad coverage obtained it might be useful to be explicit this is being done on an accurate mass instrument. Then reserve MS/MS only for the parts of the paper when metabolite IDs are being confirmed using fragment spectra.

The authors begin with a comparison of +/- mIDH and 5 v 25 mM glucose and in Fig 1 show that both manipulations driving differences in the metabolome. However, after Fig 1 the glucose perturbation is not mentioned again. It would be helpful to explain that the rest of the paper then focuses on just the WT v. mutant IDH overexpression comparison (or mIDH overexpression v. vector control cells with endogenous IDH1 levels). Also consider recapping the glucose results briefly in the discussion.

Anion-exchange chromatography mass spectrometry (IC-MS) provides in depth coverage of primary metabolic pathways revealing altered metabolism in IDH1 mutant cells.

Reviewer comments (black) and our responses (red)

Reviewer #1:

The manuscript entitled “Untargeted metabolomics by anion-exchange chromatography-mass spectrometry provides metabolic insights into gene mutations in cancer” presents a very interesting untargeted method for comprehensive and robust analysis of primary metabolic pathways. The developed methodology was deeply validated on different types of the samples including cells, tissues and bio-fluids. I do believe that this work is in high

The manuscript is nicely structured, and results are described in a logical manner. Such a large amount of results is represented in a very concise and clear way. The method was nicely described and profoundly validated and represented results only confirm its robustness. I have no objections to the substantive part of the article: introduction is properly structured and provides bases for the rest of the manuscript; materials and methods together with supplementary information provide a lot of information (although some details are missing – see comments below); results clearly illustrate the method development, validation and application process. I do believe this manuscript is high importance and interest for the readers of Communication Biology and I recommend its publication after major revision:

We thank the reviewer for this positive summary of our work.

Manuscript

- Title: In my opinion, the title of the manuscript does not fit to the content of it: From the main text and the reasoning presented in the text, I understood that the work is dedicated to the development and validation of the method for the measurement of the metabolites of primary metabolic pathways. The application of this methodology to investigate the metabolic effects of isocitrate dehydrogenase 1 (IDH1) mutations in glioblastoma cells was performed as a proof of the concept. Therefore, I would change the title, because now the title is focused only on the last part of work.

We thank the reviewer for this suggestion. We agree the title could better reflect the methodological capabilities presented and have altered the title to emphasise this and make the application more explicit. The revised title now reads:

“Anion-exchange chromatography mass spectrometry (IC-MS) provides in depth coverage of primary metabolic pathways revealing altered metabolism in IDH1 mutant cells.”

- Line 6: Please move affiliation number 2 to the next line

- Line 46: repetition of the word “challenge”

We thank the reviewer for the suggestions above; the manuscript has been updated accordingly.

- Please comment also work of Fu and colleagues on the method for Targeted Determination of Tissue Energy Status by LC-MS/MS (<https://doi.org/10.1021/acs.analchem.9b00217>)

The work by Fu and colleagues is an important reference and this has now been added to the manuscript as part of the discussion of redox metabolism and ion-pairing applications (see reference 8).

- Lines 96-100: „We used a range of metabolites with differing physicochemical properties and cell extracts for the determination of sensitivity, linearity, retention time reproducibility (robustness)

and lower limit of detection and quantification (LLOD/LLOQ). Results from the targeted validation (Supplementary Fig.1) include retention time reproducibility, linearity and the lower limit of detection (Supplementary Fig. 1b, 1c and 1d).” You determined sensitivity, linearity, retention time reproducibility (robustness) and lower limit of detection and quantification (LLOD/LLOQ), but results include only retention time reproducibility, linearity and the lower limit of detection. Why? Please comment on this and provide the missing information.

We thank the reviewer for pointing out this discrepancy and we have now removed the reference to ‘sensitivity and LLOQ’ in the manuscript for this information was not measured directly. Sensitivity is arguably most useful when comparing to another method directly which we do not do here. LLOQ can be defined as the lowest analyte concentration that provides at least a 1:10 signal to noise ratio and this has been provided for selected metabolites in previous publications (e.g. see Schweiger et al 2017) and can be estimated from the LLOD data provided here. However, given the focus on untargeted measurements in this work we opted to focus on LLOD measurements only.

- Figure 1: Please either split the figure into two or three smaller one or increase the size of the font because the graphics and axis are not readable because even using zoom it is impossible to read the legend for the PCA plots to read the labelling for the sample groups.

We have increased the image and font sizes in the panels where text was small and we have also enhanced the resolution of the images. Given these updates we have kept it as a single composite Figure but it can be further split in two if still deemed necessary.

- The caption to the figure 1 Lines 168-169: Please change the current description of HILIC and RP methods to C18-MS/MS and HILIC-MS/MS because now information about tandem analysis is missing

Upon the recommendation of another reviewer we have removed reference to the tandem mass spectrometry element of the abbreviated method: ‘IC-MS/MS’ has now become ‘IC-MS’ throughout other than when specifically referring to the tandem mass spectrometric measurements themselves. Accordingly we have kept the ‘C18-MS and HILIC-MS’ abbreviations as they are to harmonise across the manuscript.

- Please unify the nomenclature for hexose monophosphates to avoid the presence of different representations such as e.g. glucose-1P and mannose-1-P.

The nomenclature for hexose monophosphates has now been harmonised in Fig 1. and now reads ‘hexose-6P’ etc throughout.

- Line 206: For comparison of such small sample sets parametric test should not be used, even if the normality was tested. Non-parametric methods (e.g. Mann-Whitney U-test) should be used instead. In this particular case, observed differences are very strong and I do believe they will be captured also with U-test, however in case of weaker models the difference between parametric and nonparametric test might be significant.

We agree with the reviewer’s comments here and have performed non-parametric tests in addition to parametric test (normality was tested and found to be present for the t-tests). The reviewer’s assumption is correct; the reported significant changes are still significant using non-parametric tests. We have updated the manuscript to briefly reflect this information and the need for care. We have reported the parametric test results however because normality is present and the lack of change in the significance of the metabolites reported when analysed by non-parametric testing. The following text has now been added to the Fig. 1 caption.

Page 10: “Parametric and non-parametric tests were investigated to determine significance; little change in significance was found between the methods for the identified compounds reported.”

- Line 225: Please add to the correlation coefficient the symbol of absolute value $>|0.9|$

The symbol for absolute value has now been added to the text.

- Lines 244 and 248: Why two different validation methods were used? Can you justify the application of the permutation test on such a small group of samples?

The same cross-validation methods were used in both cases (leave one out cross-validation). The reviewer quite rightly implies that a small number of samples can lead to over-fitting of the data by the model and hence the importance of a permutation test, as used here, to ensure that the strength of the model determined by the cross-validation was significant (i.e $p < 0.05$). We have edited the text to make this clearer on page 9.

- Line 255: remove spare . in the Fig.. 4c

The spare '.' has been removed

- The caption to figure 2 Lines 311-312: "A white background denotes a metabolite not identified in the analysis." Do you mean identified as assigned to the true metabolites or identified as measured in the samples (metabolites not detected)? This point is not clear.

A white background refers to metabolites that were not detected, mostly commonly because they are cationic rather than anionic. The text in the caption has been updated and now reads:

"A white background denotes a metabolite not detected by the analysis"

- Figure 3: panel A-labels of the discriminating compounds are not visible – please increase the font. The same I can comment on the names of the metabolites in panel b and d. Please make adequate changes.

Figure 3 Panel A has been re-drawn to make the labelling clearer and panel B (now panel C) has been enlarged. Panel d has been removed and panel e has been increased in size and moved into a separate figure (Fig. 5).

- The caption to the Figure 3 Line 315: Please add – to the \log_{10} (it should be $-\log_{10}$...)

The minus sign has now been added to the caption for Figure 3.

- The caption to figure 3 Line 319: You stated that the hierarchical clustering was performed for the top 50 identified metabolites, why the figure shows the map for only 30 metabolites. Can you explain why?

We thank the reviewer for pointing this out. 'Top 50' in Figure 3 caption was a typo and should have read 'top 30'. However, this figure has now been removed as it was not previously referred to in the text and it was felt it simply re-visualised data already presented and therefore was not entirely necessary.

- In some parts of the manuscript (e.g. Figure 1, panel A) is stated that the RP method was used in the UPLC mode. Did you indeed performed UPLC analysis and extended 600 bars? If not, please make adequate corrections.

The RP method did use a UHPLC system for chromatography as stated. Details of this method can be found in the Methods section. The caption has been updated to include the correct abbreviation 'UHPLC' rather than 'UPLC (the latter being a commercial name)'.

- VIP score on its own cannot be used to select statistically significant variables. It must be combined with some other measures such as p-value from univariate analysis or correlation coefficient from multivariate analysis. Please check if still all selected variables can be considered as statistically significant after combining VIP with other values.

We agree with the reviewer that this was not clear in the manuscript and have re-worded the relevant section on page 9 to make clear which VIP-ranked metabolites were significant (i.e. FDR-adjusted p-value < 0.05). This is shown as tracked changes.

- Line 468: Please change rpm value to x g since rpm gives very different centrifugal force depends on the size of the rotor.

The manuscript has now been updated replacing the rpm value with 21,693 x g

- Paragraph Untargeted metabolomics method development (lines 476-486): I do not like how this part is described. Within the entire manuscript, a lot of information is provided but here very little details are given. If you decided to include in the manuscript the deplanement of the method you should provide adequate details, both about what was done and what was exact results. Stating that “for example, a MeOH precipitation was not sufficient...” (line 480) is not enough. Please, either provide information about method development, giving details about tested values etc, or remove this part from the manuscript.

We agree with the reviewer’s comment and have removed this section entirely as it simply mirrors what is already in the manuscript in greater detail, e.g. lines 80-94 in original manuscript.

- To avoid repetitions I suggest adding in the Method, a section called e.g. Instrumentation and move to it information about HPLC and MS system and stated that all the analysis were performed on..... In this way, the same information does not have to be repeated three times.

We appreciate the point made here by the reviewer but feel that on balance the information is clearer as laid out currently in separate sections rather than adding an additional section.

- In the description of MS parameters (in all three chromatographic methods) units are missing – line 499, 516 and 533). Please make adequate corrections.

Although the lines pointed out refer to gas flow rates the Q Exactive Tune software settings themselves do not relate to a specific unit of measurement and are therefore arbitrary values which is why they are reported here without units.

- Figure 1C and Figure 3D are not referred in the text – Please add adequate comments in the text or remove mentioned panel from the manuscript.

Figure 1 has now been re-organised: Figure 1c (now Figure 1e) is now referred to in the text. Figure 3d has now been removed entirely. It was felt it simply re-visualised data already presented and therefore was not necessary.

- Figure 3: Change the order of panel on the figure and their labelling to quote them in the text in order of reading. No panel A and C are quoted as first (line 218), then Panel B and (223) and finally E (line 280). As mentioned above, panel D is not quoted.

Figure 3 panel order has been re-organised and updated in the text to ensure panels are mentioned in order. All panels are now referred to in the text.

Supplementary information

- Figure 1 Panel A – The figure presented in panel A does not match the description in the caption to this figure. In the manuscript, this figure is referred (Line 90) as a comparison of filtered and nonfiltered samples, while the caption refers to the RT reproducibility. Please clarify this and make adequate changes.

The Supplementary Figure 1 caption has now been re-organised and updated in the text to match the labelled figures correctly.

- The caption to Figure 1 Panel B – I do believe that this text corresponds to the panel E on the figure. Please correct this.

Supplementary Figure 1b has now been moved to Figure 1b. The captions have been corrected and updated to match the labelled figures.

- Figure 1 Panel B – please increase the size of the font for the names of the compound on the x-axis, Panel F – increase the size of the legend of the sample groups and axis.

The Supplementary Figure 1 Panel B has been increased in size as suggested and moved to Figure 1b as it is an important finding highlighted in the main text. Figure 1d was moved to Supplementary Figure 1b. The text referring to the figure panels has also been updated accordingly.

- Figure 1 Panel D – Why the RT variability is given only for 6 out of 14 shown metabolites?

We thank the reviewer for pointing out this discrepancy which was overlooked. The missing data has now been added to the table.

- Figure 1 Panel E – please add also information about the number of features with %CV below 15 and 30 represented as a % of the total number of detected features. And comment on these results.

We have added the requested data in Supplementary Figure 1e which shows that HILIC and IC-MS both provide similar performance in terms of compound reproducibility for the samples analysed. This has been pointed out in the revised manuscript at (line 115 in original manuscript).

- The caption to Figure 1 Panel D – Change Rt to RT.

The graph that was in Figure 1, Panel D has been re-drawn to make it clearer. It has also now been moved to Supplementary Figure 1b.

- The caption to Figure 1 Panel G – The description of the table from the panel G is not clear. Please re-write it.

The caption has now been re-written and now reads: "Table showing the number of compound features measured by IC-MS in the analysis of various types of biological sample extract."

- Figure 3 Panel A: Figure shows EIC for 7 ions, but the last one (in black) is not labelled. The caption describes only 6 compounds. Can you explain this?

The additional peak has been removed so the number of labelled peaks now matches the caption.

- Figure 3 Panel A and Caption to this panel: Metabolites on the figure are labelled as A, B, C, while in the caption as 3C1, 3C2..... Please unify the labels to avoid any misunderstanding.

The caption and chromatographic peak labelling have now been unified in Supplementary Fig 3 and 4.

- Figure 5 Panel B and F: Please increase the font because now these two panels are not readable.

The font sizes have been increased for all the labels on all the panels for Figure 5.

- Figure 5 Panel D and Figure 6 Panel A: What is the meaning of colour bars on the right side of the plot? Do the colours correspond to the correlation or abundance? Please clarify this.

Coloured boxes to right hand side of each metabolite qualitatively indicate degree and direction of change in abundance between the two experimental groups. This has now been updated in the caption for the requisite panel of Figure 5 and 6.

- Figure 8-10: The codes of the compounds should be represented as names of the metabolites. This will help to read and interpreted presented results.

This has now been rectified. KEGG Pathways in Supplementary Fig. 7 & 8 have been re-drawn and are represented by metabolites names rather than KEGG codes.

- Figure 8 Panel C and Figure 10 Panel A: I do not see the reason to present pathways where single (Fig 8) or just two (fig 10) metabolites were matches.

The reviewer makes a good point. We have now removed these pathways from Figure's 8 & Figure 10.

- Figure 10: Please remove the label for panel A. It is redundant since this figure has only a single graphic.

As per recommendation above this figure has now been removed.

- Panel A and F from figure 5 are not quoted in the text. Please make adequate changes.

We thank the reviewer for pointing this out and have now added reference to both figures in the appropriate places in the text.

- Check the representation of R2 and Q2 within the manuscript and supplementary information to ensure that all are represented as superscript: R² and Q²

We thank the reviewer for pointing out this discrepancy and have now ensured all R² and Q² are superscripted in the figures and manuscript.

Reviewer #2:

The manuscript by Walsby Tickle and colleagues describes the use of anion exchange chromatography in metabolomics. They report the measurement of 431 metabolites with good coverage of mammalian central carbon metabolism. They leverage top 10 data-directed MS2 to confirm the identity of many of these metabolites and demonstrate advantages to HILIC chromatography, particularly retention time stability. The authors then apply this approach to characterize IDH1 mutations in LN18 glioblastoma cells observing increased 2-hydroxyglutarate and decreased alpha-ketoglutarate in the context of overexpression of mutant IDH1 R132H. Then, using an untargeted data analysis approach the authors also implicate changes in the lysine and tryptophan degradation pathways. The manuscript is well-written, and the experiments are clearly described and appear in a logical sequence. The RP-IP and HILIC methods used as a comparison are appropriate and informative. Metabolites that are challenging to measure using this method or give bad peak shapes have been reported. The statistical analysis methods used are appropriate.

I have the following suggestions that could strengthen an excellent manuscript: 1. Since the manuscript is centered around the benefits of anion exchange chromatography, which has not been widely employed in metabolomics, it could benefit with a more explicit introduction to how this separation differs from other commonly used separations. The methods section is comprehensive, but I think it would be of general interest to the readers to have this also outlined in the introduction.

We thank the reviewer for this useful suggestion and have added a section in the introduction to introduce IC-MS and the function of chromatographic ion-suppression which enables its application with mass spectrometry. The new section (line 62 in original manuscript) now reads:

“IC-MS is a hyphenated technique coupling conventional ion-exchange chromatography with mass spectrometry. Separation of compounds is based on ionic interaction between functional groups on a resin-based stationary phase and the analytes in the mobile phase. Elution occurs by analyte exchange with higher ion strength mobile phase ions, typically a gradient of hydroxide ions (for anion exchange) or protons (for cation exchange). Compatibility with mass spectrometry is enabled by an ion-suppressor placed between the ion chromatography system and the electrospray ion source of the mass spectrometer. In the suppressor electrochemical conversion of hydroxide ions to water molecules (in anion exchange mode) or protons to water (in cation exchange mode) provides an electrospray-compatible chromatographic eluent that can be analysed sequentially over time by the mass spectrometer. This electrochemical ion-suppressor removes mobile phase ions and oppositely charged counter ions from the sample; both processes minimise subsequent ion-suppression during electrospray ionisation in the source.”

Along the same lines: Line 84: ion-suppressor would be clarified, the typical LC-MS metabolomics audience will be familiar with the concept of ion suppression in ESI due to high abundance analytes and may be confused. Same for anion suppressor on line 86.

We agree and have included this clarification in the IC-MS introductory paragraph above and added a new figure to illustrate the chemical process (new Figure 1a).

Line 182: clarify why the technique is not suitable for zwitterionic and positive compounds. A comment on how ion exchange chromatography for the analysis of cationic species would be helpful here.

We agree and have included additional information to clarify why the technique is not suitable for zwitterion analysis. The text starting at line 182 (in the original manuscript) now reads:

“Anion-exchange chromatography retains negatively charged metabolites which are eluted from the column by an increasing hydroxide ion gradient. Online eluent suppression removes hydroxide ions from the mobile phase eluent but also all counter ions, i.e. those with a positive charge. Zwitterions, such as many amino acids, are a special case as they contain both positively and negatively charged functional groups. Most amino acids are removed from the eluent stream by this process but we show here this does not extend to all amino acids or many modified amino acids such as N-acetylated amino acids and oxo-acids (produced by amino acid transamination and which are not zwitterionic).”

2. The cell model needs to be defined more carefully: Line 104: *“Two different cell types (LN18 glioma cells genetically modified by lentiviral vectors transduction to overexpress the R132H mutant form of isocitrate dehydrogenase 1 (IDH1) and untransduced IDH1 wild-type cells).”* However, line 453: *“Immortalized LN18 (glioblastoma cell line) were obtained from ATCC (CRL-2610) and genetically modified to overexpress either wild type or mutant IDH1 by lentiviral vector transduction. Briefly cDNAs encoding to the wild type or R132H mutant sequence of IDH1 were cloned in the lentiviral transfer vectors pCC.sin.36.MCS.PPTWpre.CMV.tTA-S2tet following the methods reported in Bardella et al. 2016.”* The former indicates the control cell line is unmanipulated, the later indicates it had overexpression of WT IDH1. The experiments are valid in either case, but the mutant protein is clearly overexpressed (on top of endogenous wild type IDH1) and the distinction of whether this comparison is to endogenous levels or overexpression of the WT IDH1 is important and should be clear.

We thank the reviewer for pointing out this discrepancy. The cells were a comparison of IDH1 mutant cells and untransduced IDH1 wild-type cells. The comparison is therefore to endogenous levels. The statement in line 108 is accurate and we have now modified line 453 to reflect this. The section on cell lines in the Online Methods section now reads:

“Immortalized LN18 (glioblastoma cell line) were obtained from ATCC (CRL-2610) and genetically modified to overexpress mutant IDH1 by lentiviral vector transduction. Complementary DNA encoding to the R132H mutant sequence of IDH1 were cloned in the lentiviral transfer vectors pCC.sin.36.MCS.PPTWpre.CMV.tTA-S2tet following the methods reported in Bardella et al. 2016.27 Vectors stock and titration were prepared as described in Bardella et al., 2016.27 The IDH1 wild-type cells used for comparison were the untransduced LN18 glioblastoma cell line. All cells were grown in DMEM supplemented with 5% of fetal bovine serum (FBS) and 1% Glutamax. Cells were grown to 90% confluency.”

3. Stacked chromatogram plots could be improved. In supplementary figure 3a the black trace is missing a label, in all cases (Supplementary Fig 3a and 4a) labeling the plots with the metabolite names would help improve clarity.

The metabolites have now been labelled with metabolite names in the figure as suggested and the black trace removed for clarity.

4. The correlation plots (Fig. 3b and supplementary Fig. 5b) are dominated by correlations of +1.0 or -1.0 with 2-HG levels. This almost perfect correlation could benefit from a little more explanation – are the positively correlating metabolites in 5b in-source fragments / adducts of 2HG? Are the negatively correlating metabolites in both plots -1.0 because detection of those metabolites is lost in the mutant IDH condition?

We thank the reviewer for making this point – to answer the questions: The negative correlations in (for example Fig. 3b (now Fig. 3c)) are not lost in the mIDH1 condition but these metabolites are significantly depleted when 2-HG is enriched (see for example box plot beta-citryl glutamate in supplementary Fig. 3b). This could be due to enzyme inhibition by 2-HG but further studies are needed to confirm this as for example it could be due to loss of expression of the enzyme). For the positive correlations in Supplementary Fig. 5b (now Supplementary Fig. 5a) none of these are at the same retention time as 2-HG so are not likely to be the result of in source fragmentation for example. We have updated the text on page 8 to explain this briefly (see tracked changes).

5. Figures rely heavily on plots from MetaboAnalyst. While the data analysis methods chosen are appropriate, a little more explanation of what is being tested and displayed would be helpful to those who do not regularly use this on-line package, particularly in the figure legends.

We thank the reviewer for this important general point and have updated the figure legends (and manuscript text where relevant) with additional information about what is being tested and displayed by the various statistical methods and visualisation approaches used. See updated legends: Figures 1c, 3a, 3e, Supplementary Figures 5a, 5b and 5d.

Minor:

Why was the resolution setting for the QE HF so much lower (17,500) for IC but 70,000 for RP and IP methods?

This was not in fact the case; 70,000 resolution was used for all MS scans across all 3 methods and 12,500 resolution was used for all MS/MS scans across all 3 methods. We realise this was not clear however, and have edited the requisite methods text on page 18 accordingly.

The authors could consider replacing IC-MS/MS with IC-MS analysis (and the same for HILIC and RP methods, as it appears on lines 52/104. In my opinion the use of MS/MS in the abstract suggests a triple quad analysis, but since the authors are trying to highlight broad coverage obtained it might be useful to be explicit this is being done on an accurate mass instrument. Then reserve MS/MS only for the parts of the paper when metabolite IDs are being confirmed using fragment spectra.

We agree with the reviewer about the use of the terms IC-MS and IC-MS/MS. The manuscript has been updated throughout to represent the hyphenated technique at IC-MS accordingly.

The authors begin with a comparison of +/- mIDH and 5 v 25 mM glucose and in Fig 1 show that both manipulations driving differences in the metabolome. However, after Fig 1 the glucose perturbation is not mentioned again. It would be helpful to explain that the rest of the paper then focuses on just the WT v. mutant IDH overexpression comparison (or mIDH overexpression v. vector control cells with endogenous IDH1 levels). Also consider recapping the glucose results briefly in the discussion.

We thank the reviewer for these comments: We have updated the text to make clearer the distinction between glucose media concentrations used in the study. The paragraph starting at line 203 now reads:

“To investigate the links between IDH mutation status and perturbed metabolism in more detail we cultured six IDH1 (R132H) mutant replicate dishes of cells and six IDH1 wild-type LN18 glioblastoma replicate dishes of cells both in the same media containing 5mM glucose (5mM is closer to the glucose concentration in the human brain than standard 25mM DMEM). Metabolite extracts from each dish were analysed using untargeted IC-MS and univariate statistical analysis (non-parametric t-tests) along with fold-change (FC) for each compound feature were calculated.”

END

REVIEWERS' COMMENTS:

Reviewer # 1 (Remarks to the Author)

"The vast majority of comments have been taken into account. I am happy with changes made in the manuscript. I would like to comment some of the answers from the authors:

"We agree with the reviewer's comments here and have performed non-parametric tests in addition to parametric test (normality was tested and found to be present for the t-tests). The reviewer's assumption is correct; the reported significant changes are still significant using non-parametric tests. We have updated the manuscript to briefly reflect this information and the need for care. We have reported the parametric test results however because normality is present and the lack of change in the significance of the metabolites reported when analysed by non-parametric testing. The following text has now been added to the Fig. 1 caption." It is a great information that there are no significant differences between two types of the statistics used. However, for the future please avoid using parametric test for such small sample size. Even if the normality was tested, and the results showed the normality, the results of such testing are not robust. Rested group is too small to measure in robust way the normality.

"The same cross-validation methods were used in both cases (leave one out cross-validation). The reviewer quite rightly implies that a small number of samples can lead to over-fitting of the data by the model and hence the importance of a permutation test, as used here, to ensure that the strength of the model determined by the cross-validation was significant (i.e $p < 0.05$). We have edited the text to make this clearer on page 9." But from the text I cannot get this idea, just the opposite that in the first case authors applied permutation and in the second both, permutation and cross validation (see the comment below for the lines 286-293). Please, clarify this. There are still few issues that have to be fixed. First of all profound English revision is required. Below I listed some other issues:

Line 64 – please change „mobiles phase“ to “mobile phases”

Line 123 – “Fig. 1c additional plots” – what does additional plots mean?

Section “Extending ionic metabolome coverage”: Authors stated: “This list extends considerably what so far been demonstrated for the analysis of metabolites using anion IC-MS”. Please add a reference to prove the truth of this statement.

Lines 126-129: Why RSD 15% was used as a criterion to justify and compare the reproducibility?

Figure 2: In the footage of the figure red boxes are described as “no standard available” and there is also a symbol of □ indicating metabolites annotated based on the HMDB match due to the lack of the standard. What is the difference between this two categories? Especially that not all boxes with □ symbol have red frame. Please explain this and clarify this in the manuscript.

Lines 286-293: “(Supplementary Fig. 5c, red star; $R^2=0.96$, $Q^2=0.94$, Accuracy = 1.0;

Supplementary Fig. 5e shows 1000 permutations were used to ensure the model did not over-fit the data (p -value < 0.001). Interestingly when we removed 2-HG from the dataset, and remodelled the data, a minimum of only 8 new compound features were required to differentiate IDH1 mutant and IDH1 wild-type cells (Fig. 4a shows VIP plot). Model strength and accuracy was high ($R^2=0.99$, $Q^2=0.96$ and Accuracy > 0.99) using the same ‘leave one out’ cross-validation approach (Supplementary Fig. 5d) although the model no longer passed a permutation test (1000 permutations) showing the importance of 2-HG to the model's strength.” First authors quote permutation and then bring “leave one out” method as “the same ‘leave one out’ cross-validation approach”. It is true, cross validation and permutation are validation methods but they are two different approaches. Moreover, permutation is not an adequate method for such small sample size, and correct approach is cross-validation.

Figure 3 panels b and d: Please unify the way to display the numbers and unify the number of significant figures.

Figure 3 panel c: Please edit the figure to replace the names of the metabolites because some of them

are not fully displayed.

Figure 4 panel a: Please edit the figure to replace the names of the metabolites because some of them are not fully displayed.

Figure 4: please unify the names of metabolites between different panels to avoid e.g. glutamic acid and glutamate.

Figure 4: From the footage of this figure I understood that the box plots were selected based on the VIP score. The list of top 30 scores is displayed in panel A. But this panel is not showing, e.g. Glutathione or Phosphate but these metabolites are shown with the boxplots. Can you please explain this?

Figures: I have a sensation that some of the description in footage of figures are too long. In my opinion footage should provide information necessary to read and interpret the figures but not the interpretation itself. E.g. in footage of figure 3 we can find "This suggests their respective endogenous metabolism may be significantly different." This should be in the text but not in the footage of the figure itself.

Materials and methods: Please unify the way to present units: either ml/min or ml min⁻¹.

In my opinion in the results and discussion section there are many repetitions from the materials and method sections. For example, once the details about the number of permutations were given in Materials and methods it is not necessary to repeat this in results. Just the information about permutation as a validation method is enough. The same about the cut-offs for p value, fold change etc. Please revise the results and remove the repetitions and redundancies.

Reviewer #2 (Remarks to the Author):

The authors have satisfactorily addressed my comments. I recommend this excellent manuscript for publication in Nature Communications Biology with no further changes.

Reviewer #2 (Remarks to the Author): (our responses in blue)

The authors have satisfactorily addressed my comments. I recommend this excellent manuscript for publication in Nature Communications Biology with no further changes.

We thank the reviewer for their generous comments and time.

Reviewer 1

The vast majority of comments have been taken into account. I am happy with changes made in the manuscript. I would like to comment some of the answers from the authors:

We are pleased to see that the reviewer is happy with the changes made to the manuscript and we answer the additional contextual comments below.

“We agree with the reviewer’s comments here and have performed non-parametric tests in addition to parametric test (normality was tested and found to be present for the t-tests). The reviewer’s assumption is correct; the reported significant changes are still significant using non-parametric tests. We have updated the manuscript to briefly reflect this information and the need for care. We have reported the parametric test results however because normality is present and the lack of change in the significance of the metabolites reported when analysed by non-parametric testing. The following text has now been added to the Fig. 1 caption.” It is a great information that there are no significant differences between two types of the statistics used. However, for the future please avoid using parametric test for such small sample size. Even if the normality was testes, and the results showed the normality, the results of such testing are not robust. Rested group is too small to measure in robust way the normality.

We thank the reviewer for their comments

“The same cross-validation methods were used in both cases (leave one out cross-validation). The reviewer quite rightly implies that a small number of samples can lead to over-fitting of the data by the model and hence the importance of a permutation test, as used here, to ensure that the strength of the model determined by the cross-validation was significant (i.e $p < 0.05$). We have edited the text to make this clearer on page 9.” But from the text I cannot get this idea, just the opposite that in the first case authors applied permutation and in the second both, permutation and cross validation (see the comment below for the lines 286-293). Please, clarify this.

We thank the reviewer for these comments. The permutation testing and cross validation were conducted both before and after removal of 2-HG from the dataset and the results of these test statistics are provided in Supplementary Figure 5c, 5d and 5e (with the exception of the permutation test statistic after removal of 2-HG as it was not significant, as stated in the text). We believe this was communicated in the manuscript but given the reviewer’s request for further clarity, and further comments below on Lines 286-293 specifically below, we have removed reference to the permutation tests in the manuscript given the reviewers remarks that cross-validation is a more appropriate validation method in this case, given the numbers of samples used in the study. The relevant section now reads:

(Supplementary Fig. 5c shows results of the cross-validation test ($R^2=0.96$, $Q^2=0.94$, Accuracy = 1.0) used to check the model was not over-fitting the data. Interestingly when we removed

2-HG from the dataset, and remodelled the data, a minimum of only 8 new compound features were required to differentiate IDH1 mutant and IDH1 wild-type cells (Fig. 4a shows VIP plot). The model strength and accuracy was still high ($R^2=0.99$, $Q^2=0.96$ and Accuracy >0.99), using the same 'leave one out' cross-validation approach (Supplementary Fig. 5d).

There are still few issues that have to be fixed. First of all profound English revision is required. Below I listed some other issues:

Line 64 – please change „mobiles phase” to “mobile phases” This has now been corrected.

Line 123 – “Fig. 1c additional plots” – what does additional plots mean? This simply referred to the other two PCA plots in Fig 1c ‘HILIC-MS and ‘UHPLC-MS’ but is redundant and has been removed.

Section “Extending ionic metabolome coverage”: Authors stated: “This list extends considerably what so far been demonstrated for the analysis of metabolites using anion IC-MS”. Please add a reference to prove the truth of this statement. Reference have now been added.

Lines 126-129: Why RSD 15% was used as a criterion to justify and compare the reproducibility? This was arbitrary given we reported both %CV <15 and <30. We have updated to text to comment on similarities between IC-MS and HILIC-MS at both thresholds now.

Figure 2: In the footage of the figure red boxes are described as “no standard available” and there is also a symbol of □ indicating metabolites annotated based on the HMDB match due to the lack of the standard. What is the difference between this two categories? Especially that not all boxes with □ symbol have red frame. Please explain this and clarify this in the manuscript. Red borders with a dagger indicate we were able make a putative ID based on HMDB. Red borders without dagger means we found no evidence for the metabolite in the sample analysis. All metabolites marked with a dagger should be in a red frame, thanks to the reviewer for pointing this out. The figure has been updated.

Lines 286-293: “(Supplementary Fig. 5c, red star; $R_2=0.96$, $Q_2=0.94$, Accuracy = 1.0; Supplementary Fig. 5e shows 1000 permutations were used to ensure the model did not over-fit the data (p -value <0.001)). Interestingly when we removed 2-HG from the dataset, and remodelled the data, a minimum of only 8 new compound features were required to differentiate IDH1 mutant and IDH1 wild-type cells (Fig. 4a shows VIP plot). Model strength and accuracy was high ($R_2=0.99$, $Q_2=0.96$ and Accuracy >0.99) using the same 'leave one out' cross-validation approach (Supplementary Fig. 5d) although the model no longer passed a permutation test (1000 permutations) showing the importance of 2-HG to the model’s strength.”

First authors quote permutation and then bring “leave one out” method as “the same 'leave one out' cross-validation approach”. It is true, cross validation and permutation are validation methods but they are two different approaches. Moreover, permutation is not an adequate method for such small sample size, and correct approach is cross-validation. We thank the reviewer for these comments. The permutation testing and cross validation were conducted both before and after removal of 2-HG from the dataset and the results of these test statistics are provided in Supplementary Figure 5c, 5d and 5e (with the exception of the permutation test statistic after removal of 2-HG as it was not significant, as stated in the text). We believe this was communicated in the manuscript but given the reviewer’s request for further clarity, and further comments below

on Lines 286-293 specifically below, we have removed reference to the permutation tests in the manuscript given the reviewers remarks that cross-validation is a more appropriate validation method in this case, given the numbers of samples used in the study.

Figure 3 panels b and d: Please unify the way to display the numbers and unify the number of significant figures. This has now been done.

Figure 3 panel c: Please edit the figure to replace the names of the metabolites because some of them are not fully displayed. This has now been done.

Figure 4 panel a: Please edit the figure to replace the names of the metabolites because some of them are not fully displayed. This has now been done.

Figure 4: please unify the names of metabolites between different panels to avoid e.g. glutamic acid and glutamate. This has now been done (Fig 4a and 4c edited to harmonise naming to anion form)

Figure 4: From the footage of this figure I understood that the box plots were selected based on the VIP score. The list of top 30 scores is displayed in panel A. But this panel is not showing, e.g. Glutathione or Phosphate but these metabolites are shown with the boxplots. Can you please explain this?

We thank the reviewer for pointing out this discrepancy. The VIP plot was correct but some of the identified metabolites had not been annotated on the plot. These have now been included and the box plots match the identifications and compound feature names in the VIP plot.

Figures: I have a sensation that some of the description in footage of figures are too long. In my opinion footage should provide information necessary to read and interpret the figures but not the interpretation itself. E.g. in footage of figure 3 we can find "This suggests their respective endogenous metabolism may be significantly different." This should be in the text but not in the footage of the figure itself.

All captions are within the text limits set by NCB, however, we agree that captions should provide information necessary to read and interpret the figures but not discuss their interpretation. Accordingly we have removed some text from Figure 1 and Figure 3.

Materials and methods: Please unify the way to present units: either ml/min or ml min⁻¹. All have now been updated to 'ml/min'

In my opinion in the results and discussion section there are many repetitions from the materials and method sections. For example, once the details about the number of permutations were given in Materials and methods it is not necessary to repeat this in results. Just the information about permutation as a validation method is enough. The same about the cut-offs for p value, fold change etc. Please revise the results and remove the repetitions and redundancies.

We thank the reviewer for this general point and have revised the manuscript accordingly where we think it appropriate. With respect to significance thresholds and fold-changes we have kept these in all figures captions (and a small number of instances in the main text) as we think this can be helpful to readers.